# GEM-FI: Gated Evidential Mixtures with Fisher Modulation

**Marco Mustafa Mohammed** [1]   **Fatemeh Daneshfar** [1]   **Pietro Liò** [2]

## Abstract

Evidential Deep Learning (EDL) enables single-pass uncertainty estimation by predicting Dirichlet evidence, but it can remain overconfident and poorly calibrated, and it often fails to represent multi-modal epistemic uncertainty. We introduce **G**ated **E**vidential **M**ixtures (GEM), a family of models that learns an in-model energy signal and uses it to gate evidential outputs end-to-end in a distance-informed manner. GEM-CORE learns a feature-level energy and maps it to a bounded gate that smoothly suppresses evidence when support is low. To capture epistemic multi-modality without multi-pass ensembling, GEM-MIX adds a lightweight mixture of evidential heads with learned routing weights while preserving single-pass inference. Finally, GEM-FI stabilizes mixture allocations via a Fisher-informed regularizer, reducing head collapse and producing smoother boundary uncertainty. Across image classification and OOD detection benchmarks, GEM improves calibration and ID/OOD separation with single-pass inference. On CIFAR-10, GEM-FI vs. DAEDL improves Acc. from 91.11 to 93.75 (+2.64 pp), reduces Brier$\times 100$ from 14.27 to 6.81 ($-7.46$), and also improves misclassification-detection (AUPR) from 99.08 to 99.94 (+0.86). For epistemic OOD detection, GEM-FI achieves AUPR/AUROC of 92.59/95.09 on CIFAR-10$\rightarrow$SVHN and 90.20/89.06 on CIFAR-10$\rightarrow$CIFAR-100 (vs. 85.54/89.30 and 88.19/86.10 for DAEDL).

## 1. Introduction

Reliable predictive uncertainty is essential when models operate beyond their training distribution or in safety-critical settings (Ovadia et al., 2019). A calibrated model should be confident only on well-supported regions and defer elsewhere. Bayesian neural networks (BNNs) offer a principled route by placing distributions over weights (Blundell et al., 2015), but deployments often face prohibitive training or inference costs. Popular approximations such as Monte Carlo dropout (MC-DROPOUT) (Gal & Ghahramani, 2016) and deep ensembles (Lakshminarayanan et al., 2017) improve robustness, yet require multiple forward passes, which can be at odds with tight latency or energy budgets.

EDL (Sensoy et al., 2018) provides a single-pass alternative by predicting Dirichlet parameters and interpreting their concentration as "evidence." While competitive on in-distribution (ID), networks may still be overconfident under distribution shift or for out-of-distribution (OOD) inputs (Ovadia et al., 2019; Minderer et al., 2021). Density-aware variants (e.g., Density-Aware Evidential Deep Learning (DAEDL) (Yoon & Kim, 2024)) rescale evidential outputs using an offline feature-space likelihood, such as Gaussian Discriminant Analysis (GDA) (Murphy, 2012). This improves calibration, but keeps the density cue *static* and *decoupled* from end-to-end learning. This decoupling creates several gaps that motivate our approach: (i) the offline density is not optimized jointly with the evidential mechanism, so the network cannot learn to *shape* evidence where support is weak; (ii) the density surrogate is brittle to representation shift: when features drift, the pre-fit likelihood can systematically mis-rank near-boundary or near-OOD inputs; and (iii) DAEDL does not address *epistemic multimodality* near complex class boundaries, where a single evidential head may collapse to overconfident allocations unless explicitly regularized.

Energy-based views (Liu et al., 2020) often yield stronger ID/OOD separability than softmax confidence, yet they are typically used *post hoc*: thresholds and temperatures are tuned after training and do not intervene where evidential confidence is *produced* (Guo et al., 2017). As a result, energy signals rarely enforce *local smoothness* (e.g., via Lipschitz-oriented regularization such as Parseval constraints (Cissé et al., 2017) or spectral normalization (Miyato et al., 2018)) or *distance-aware monotonicity* of evidence during learning, and can exhibit dataset-specific sensitivity (Ovadia et al., 2019; Minderer et al., 2021). More broadly, many single-pass pipelines either rely on static, decoupled density surrogates (e.g., DAEDL (Yoon & Kim,

[1]Department of Computer Engineering, Faculty of Engineering, University of Kurdistan, Sanandaj, Iran [2]Department of Computer Science and Technology, University of Cambridge, Cambridge, United Kingdom. Correspondence to: Fatemeh Daneshfar <Daneshfarshadi@gmail.com>.

*Proceedings of the $43^{rd}$ International Conference on Machine Learning*, Seoul, South Korea. PMLR 306, 2026. Copyright 2026 by the author(s).

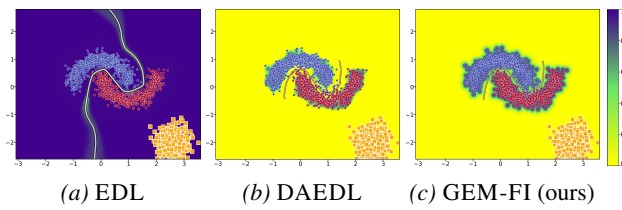

| *(a) EDL* | *(b) DAEDL* | *(c) GEM-FI (ours)* |

*Figure 1.* Two-moons setup with an additional OOD cluster. Panels show predictive entropy (brighter = higher uncertainty).

2024)) or apply post hoc score adjustments (e.g., temperature scaling (TS), energy scoring) (Guo et al., 2017; Romero et al., 2024). Consequently, there is a need for an in-model, learnable support signal that (i) directly gates evidential outputs, (ii) preserves single-pass inference, and (iii) captures epistemic multi-modality without multi-pass ensembles.

This paper asks a simple question: Can we integrate a data-dependent notion of representation "support" directly into the evidential mechanism, while retaining single-pass inference? We answer in the affirmative with GEM. GEM-CORE learns a feature-level energy and maps it to a bounded in-model gate that smoothly scales evidence. The mapping from energy to the final integration gate is learned; empirically, off-support inputs yield smaller gates and thus more conservative predictions. To capture multi-modal epistemic structure near complex decision boundaries without multi-pass ensembling, Mixture of Beliefs (GEM-MIX) augments EDL with a single-pass mixture of evidential heads and learned mixture weights. Finally, GEM-FI introduces an Fisher-informed (FI) regularizer that stabilizes allocations and discourages head collapse, yielding smoother boundary uncertainty and stronger suppression off-support. On a synthetic two-moons setup, Figure 1 compares EDL, DAEDL, and GEM-FI on the same data. EDL (Fig. 1a) concentrates uncertainty in a narrow band near the decision boundary while remaining overconfident far from support. DAEDL (Fig. 1b) improves distance awareness and calibration, but its density cue is static and decoupled from end-to-end learning, and it can still underestimate uncertainty near the curved boundary and in parts of the OOD region. In contrast, GEM-FI (Fig. 1c) yields smoother uncertainty near decision boundaries and more consistently lower confidence on OOD inputs. In a corresponding one-dimensional example, Figure 2 contrasts a single-head DAEDL model (Fig. 2a) with the FI-regularized multi-head GEM-FI (Fig. 2b), illustrating how our mixture retains uncertainty across modes instead of collapsing to overconfident allocations.

Our design follows two principles. (i) **Distance-informed confidence:** we learn a representation-level energy $E(x)$ and pass it through a bounded gate that directly scales evidential outputs, so higher energy yields a smaller gate and more conservative evidence. This creates a smooth link between feature-space support and confidence, reduces abrupt overconfidence under shift, and keeps the support signal

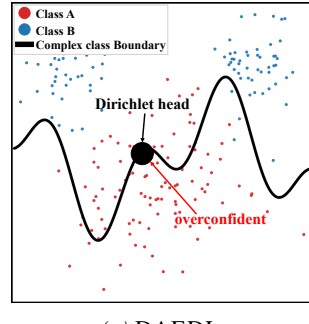 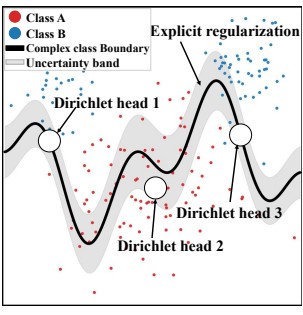

| *(a) DAEDL* | *(b) GEM-FI (ours)* |

*Figure 2.* One-dimensional toy example in a non-convex boundary region. (a) DAEDL single-head model: tends to miss epistemic multi-modality and can yield overconfident allocations. (b) FI-regularized multi-head GEM-FI in the same region: retains uncertainty across modes and avoids overconfident collapse.

learnable and end-to-end (rather than a static, offline density). (ii) **Single-pass epistemics:** we recover ensemble-like diversity with a lightweight mixture of evidential heads trained jointly on a shared backbone. Learned mixture weights provide soft specialization near complex decision boundaries, while FI stabilization discourages head dominance and improves calibration and OOD separability—all with single-pass inference and modest overhead. Empirically, across standard image-classification and OOD detection benchmarks, GEM improves calibration and strengthens ID/OOD separation relative to EDL and strong density-aware baselines.

Our main contributions are summarized as follows:

- We learn a feature-level energy and map it to a bounded, in-model gate that directly modulates Dirichlet evidence, reducing overconfidence for atypical features while preserving confident ID predictions.

- We introduce a lightweight mixture of evidential heads with learned routing weights, capturing multi-modal epistemic structure without ensembles or extra forward passes.

- We add a FI-informed regularizer to stabilize mixture allocations and prevent head collapse, yielding smoother boundary uncertainty.

- We demonstrate improvements in calibration and OOD separation on standard benchmarks, and provide ablations isolating the roles of the energy gate, mixture size, and FI regularization.

We defer the related-work discussion and a compact comparison with the closest single-pass evidential and density-aware methods to Appendix C (Table 5).

**Conflict of Interest Disclosure.** The authors declare no financial conflicts of interest related to this work.

## 2. Method

Figure 3 summarizes the architectures considered in this work. Figure 3a presents the density-aware DAEDL base-

line, while Figure 3b illustrates the proposed GEM-FI architecture and its main components. The added modules in GEM—the energy head, integration gate, router, and evidential heads—are lightweight attachments on top of a backbone that produces fixed-dimensional features, so the design is compatible with standard CNN or Transformer-style feature extractors with minimal architectural assumptions. GEM-CORE augments a spectrally normalized backbone with a learned feature-level energy and a bounded gate that modulates the predictive distribution via probability-space gating (Sec. 2.1). GEM-MIX introduces a single-pass mixture of evidential heads with learned mixing weights (Sec. 2.2). GEM-FI adds an FI-informed regularizer, together with FI-based modulation of mixture weights, to stabilize allocations (Sec. 2.3).

**Notation.** Let $x \in \mathcal{X}$ and $y \in \{1, \ldots, C\}$ denote an input and its class label, where $C$ is the number of classes. A spectrally normalized backbone $f_\theta : \mathcal{X} \to \mathbb{R}^d$ produces features $z = f_\theta(x)$.

**Evidential parameterization.** An evidential head $g_\phi : \mathbb{R}^d \to \mathbb{R}^C$ maps these features to logits:

$$u_{1:C}(x) = g_\phi(z). \tag{1}$$

Dirichlet distributions are denoted by $\mathrm{Dir}(\alpha)$ for $\alpha \in \mathbb{R}^C_{>0}$, with total concentration $\alpha_0 = \sum_c \alpha_c$ and expectations $\mathbb{E}_\alpha[\cdot]$ taken with respect to $\mathrm{Dir}(\alpha)$. Throughout, we parameterize $\alpha$ directly from (clipped) logits and use a small $\epsilon > 0$ for numerical stability:

$$\begin{aligned} \tilde{u}_c(x) &= \mathrm{clip}\big(u_c(x), -\tau, \tau\big), \\ \alpha_c(x) &= \exp\big(\tilde{u}_c(x)\big) + \epsilon, \qquad \epsilon = 10^{-8}. \end{aligned} \tag{2}$$

Unless stated otherwise, $u_c(x)$ and $\alpha_c(x)$ denote the *single-head* logits and concentrations used by EDL/GEM-CORE. For mixture models, we write $u_c^{(k)}(x)$, $\alpha_c^{(k)}(x)$, and $p_c^{(k)}(x)$ for the corresponding quantities of head $k$. We reserve $\hat{s}(x)$ for the intermediate scalar energy-derived gate signal, $s_c(x)$ for the final class-wise integration gates, and $\pi_k(x)$ for router-produced mixture weights.

**DAEDL baseline.** For reference, DAEDL (Yoon & Kim, 2024) keeps the same backbone $f_\theta$ and evidential head $g_\phi$ as above, and therefore uses the same logits $u_c(x)$. It additionally fits an *offline* feature-space density model (e.g., class-conditional GDA) on the features $z = f_\theta(x)$, and uses its normalized likelihood to modulate these logits (Figure 3a). The Dirichlet parameters and predictive mean in DAEDL are:

$$\begin{aligned} \alpha_c^{\mathrm{DAEDL}}(x) &= \exp\big(\lambda(x)\, u_c(x)\big), \\ p_c^{\mathrm{DAEDL}}(x) &= \frac{\alpha_c^{\mathrm{DAEDL}}(x)}{\sum_j \alpha_j^{\mathrm{DAEDL}}(x)}. \end{aligned} \tag{3}$$

The first line scales the logit $u_c(x)$ by the density-dependent factor $\lambda(x)$ to form the Dirichlet concentration $\alpha_c^{\mathrm{DAEDL}}(x)$, and the second line normalizes these concentrations to obtain the predictive probability $p_c^{\mathrm{DAEDL}}(x)$. The density term

$\lambda(x)$ is computed once from the offline surrogate $q(z)$ and kept fixed during training; in contrast, our GEM-CORE uses a learnable, in-model gate $s(x)$ that is trained jointly with the backbone and evidential heads.

### 2.1. GEM-CORE: Energy-to-Gate Evidential Learning Density Scaling.

In addition to the learned gate, we employ a lightweight density scaler $\rho(z)$ to modulate evidential concentrations based on feature density. We estimate $\rho(z) = \sigma(\log p(z))^\gamma$, where $p(z)$ is the log-likelihood from a Gaussian Mixture Model (GMM) fit to ID training features, $\sigma$ is the sigmoid function, and $\gamma = 1.2$ is a fixed exponent. This density score is used purely as a multiplicative scaler on the evidence: $\alpha_c(x) = \rho(z) \cdot \exp(\tilde{u}_c(x)) + \epsilon$. In mixture variants, the same shared feature-density score is applied per head before normalization, yielding $\alpha_c^{(k)}(x) = \rho(z) \cdot \exp(\mathrm{clip}(u_c^{(k)}(x), -\tau, \tau)) + \epsilon$ unless otherwise noted. This supplementary component acts as a "hard" safety guardrail to suppress evidence in regions of extremely low density, while the learned gate $s(x)$ remains the primary, task-adaptive support signal responsible for fine-grained modulation.

**Energy convention and gate direction.** We define energy $E(x)$ such that *higher energy corresponds to lower representation-level support* (anti-correlated with feature density). The intermediate scalar $\hat{s}(x) = \sigma(E(x)) \in (0, 1)$ increases with energy by construction. However, the integration gate network $G_\eta$ takes $[z, \hat{s}(x)]$ as input and learns to output per-class gates $s(x) \in [s_{\min}, s_{\max}]^C$. Crucially, $G_\eta$ can learn either positive or negative correlation with $\hat{s}$; empirically, we observe an *inverse-like mapping*: higher $\hat{s}$ (indicating lower support) leads to smaller final gates $s(x)$, thereby suppressing evidence for OOD inputs. This correspondence is a *learned outcome* of end-to-end training, not an architectural constraint.

GEM-CORE learns a feature-level energy $E_\psi : \mathbb{R}^d \to \mathbb{R}$ and maps it to a bounded (class-wise) gate $s(x) \in [s_{\min}, s_{\max}]^C \subset (0, 1)^C$ that directly modulates class probabilities via probability-space gating. $E_\psi$ is a lightweight MLP on $z$. The scalar energy $E(x)$ is first squashed with a sigmoid to obtain an intermediate scalar gate $\hat{s}(x) = \sigma(E(x)) \in (0, 1)$. This scalar is then concatenated with $z$ and fed into a small "integration gate" network $G_\eta$ that outputs per-class gates, with:

$$\begin{aligned} \tilde{u}_c(x) &= \mathrm{clip}\big(u_c(x), -\tau, \tau\big), \\ \alpha_c(x) &= \rho(z) \cdot \exp(\tilde{u}_c(x)) + \epsilon, \\ p_c(x) &= \frac{\alpha_c(x)}{\alpha_0(x)}. \end{aligned} \tag{4}$$

**Probability-space gating (implementation).** The final class-wise gate is denoted by $s(x) = (s_1(x), \ldots, s_C(x))$ to distinguish it from the intermediate scalar signal $\hat{s}(x)$. This per-class gate is applied to the predictive distribution in probability space and then renormalized:

$$\hat{p}(x) = \frac{p(x) \odot s(x)}{\mathbf{1}^\top\big(p(x) \odot s(x)\big)}. \tag{5}$$

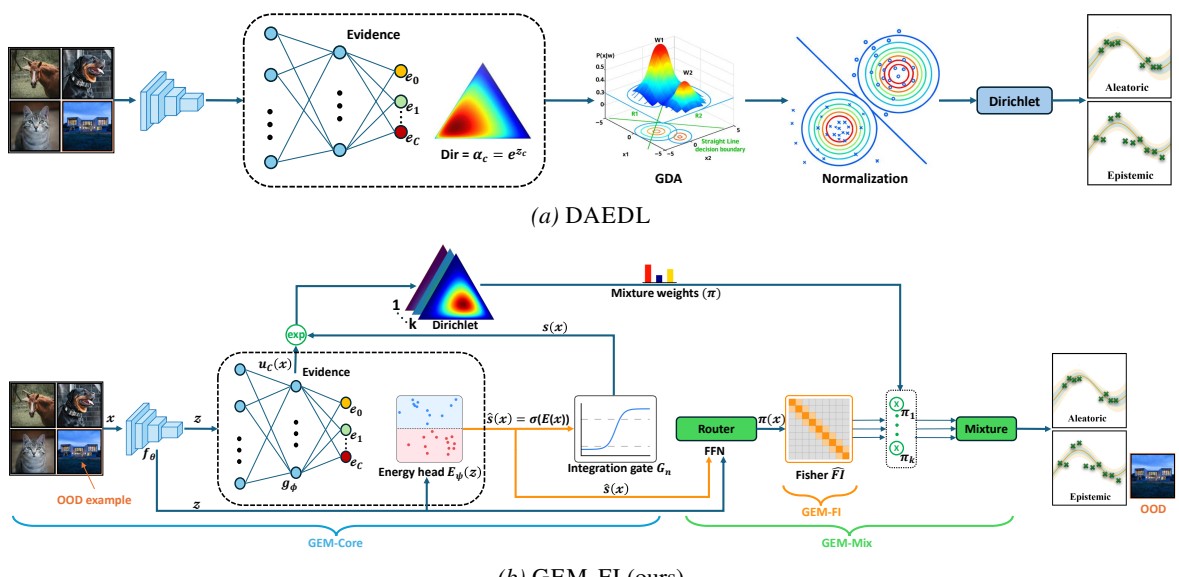

*(a)* DAEDL

*(b)* GEM-FI (ours)

*Figure 3.* Architecture of the proposed method. (a) DAEDL: a spectrally normalized backbone with a single evidential head that outputs Dirichlet evidence, augmented with an offline feature-space density model (GDA) whose normalized likelihood rescales evidential outputs before computing uncertainty. (b) GEM-FI: extends the same backbone with an energy head $E_\psi$ that maps features $z$ to a scalar energy $E(x)$ and a bounded class-wise gate $s(x)$ (GEM-CORE), and adds a router that produces mixture weights over multiple Dirichlet heads together with a FI-based regularizer.

In this block, $p(x)$ denotes the predictive mean in probability space (either a single-head evidential predictive mean or the mixture predictive mean). The per-class gate $s(x)$ is applied multiplicatively to $p(x)$ and the result is renormalized to obtain $\hat{p}(x)$ in (5), matching the implementation (probability-level gating). Training minimizes a standard evidential target-matching loss with a Kullback–Leibler (KL) prior to the uniform Dirichlet:

$$\mathcal{L}_{\text{core}} = \mathbb{E}_{(x,y)}\Big[\big\| e_y - \hat{p}(x) \big\|_2^2 + \lambda_{\text{KL}} \, \text{KL}\big[\text{Dir}(\alpha(x)) \,\|\, \text{Dir}(\mathbf{1})\big]\Big]. \quad (6)$$

Here, the core loss combines a squared error term that matches the gated predictive mean $\hat{p}(x)$ to the one-hot label $e_y$ with a KL regularizer that keeps the concentration vector $\alpha$ close to the non-informative prior $\text{Dir}(\mathbf{1})$. Since $\hat{p}(x)$ depends on both the energy head $E_\psi$ (via $s(x) = G_\eta(E_\psi(z))$) and the gate network $G_\eta$, gradients flow back through both components during training, enabling end-to-end learning of the density-aware gating mechanism.

**Inference (single-pass, no gradients).** At inference time, the model performs a single forward pass through the frozen network: we compute features $z = f_\theta(x)$, energy $E(x)$, gates $s(x)$, and mixture weights $\pi(x)$ via the router–all without any gradient computation. The predictive mean is $\mathbb{E}_\alpha[\pi]$; proxies include $\alpha_0$ (epistemic), $\max_c \mathbb{E}_\alpha[\pi_c]$ (aleatoric), entropy/MI, and an energy-derived score (reported as an auxiliary single-pass baseline for shift/OOD scoring; not used beyond the gating pipeline). Importantly, the FI-based modulation of mixture weights (17) and the FI regularizer (13) are applied only during training; at inference, mixture weights are computed directly from the router output. In our implementation, the final $\tanh$ nonlinearity on the en-

ergy head output is *optional* and disabled by default; we found that removing it (i.e., using an identity mapping) improves OOD separation, particularly when combined with energy-based-model (EBM) negative sampling using Virtual Outlier Synthesis (VOS). When $\tanh$ is enabled, a mild "desaturation" can be applied at evaluation time by scaling the pre-activation by $0.5$ to avoid hard saturation.

**Complexity.** GEM-CORE adds only the energy head $E_\psi$ and the integration gate $G_\eta$; inference remains single-pass with no gradient computation required.

### 2.2. GEM-MIX: Mixture of Beliefs (Single-Pass)

To capture multi-modal epistemic structure near complex decision boundaries without multi-pass ensembling, GEM-MIX extends GEM-CORE with $K$ evidential heads $\{g_{\phi^{(k)}}\}_{k=1}^K$ that share backbone features $z = f_\theta(x)$. Each head outputs class-wise logits $u^{(k)}(x) \in \mathbb{R}^C$, which are mapped to Dirichlet concentrations as:

$$\alpha^{(k)}(x) = \exp\big(\text{clip}(u^{(k)}(x), -\tau, \tau)\big) + \varepsilon, \qquad \varepsilon = 10^{-8}. \quad (7)$$

For readability, we write $\alpha^{(k)}(x)$ for the full concentration vector of head $k$ and $\alpha_c^{(k)}(x)$ for its class-$c$ entry; likewise, $p^{(k)}(x)$ denotes the per-head predictive mean vector and $p_c^{(k)}(x)$ its class-$c$ component. The predictive mean for head $k$ is

$$p_c^{(k)}(x) = \frac{\alpha_c^{(k)}(x)}{\sum_j \alpha_j^{(k)}(x)}. \quad (8)$$

A learnable router $h_\omega$ takes the shared features along with the scalar energy gate and produces mixture weights:

$$\pi(x) = \text{softmax}\big(h_\omega([z, \hat{s}(x)])\big) \in \Delta^{K-1}. \quad (9)$$

The mixture predictive mean (before per-class probability gating) is then

$$p_{\text{mix}}(y=c \mid x) = \sum_{k=1}^{K} \pi_k(x)\, p_c^{(k)}(x), \qquad (10)$$

$$\alpha_{0,\text{mix}}(x) = \sum_{k=1}^{K} \pi_k(x)\, \alpha_0^{(k)}(x), \qquad (11)$$

where $\alpha_0^{(k)}(x) = \sum_c \alpha_c^{(k)}(x)$. The final predictive distribution is obtained by applying the shared per-class gate in probability space and renormalizing as in (5): $\hat{p}(x) = \text{Normalize}\big(p_{\text{mix}}(x) \odot s(x)\big)$.

We train GEM-MIX using a negative log-likelihood term on $\hat{p}$ together with per-head KL priors:

$$\mathcal{L}_{\text{mix}} = \mathbb{E}_{(x,y)}\Big[ -\log \hat{p}_y(x) \\ + \lambda_{\text{KL}} \sum_{k=1}^{K} \pi_k(x)\, \text{KL}\big[\text{Dir}(\alpha^{(k)}(x)) \,\|\, \text{Dir}(\mathbf{1})\big]\Big]. \qquad (12)$$

### 2.3. GEM-FI: FI-Informed Regularization and Modulation

To further stabilize mixture behavior and discourage head collapse, GEM-FI augments GEM-MIX with an FI-informed regularizer and an FI-based modulation of the mixture weights.

**FI proxy computation.** We compute a lightweight per-head proxy $\widehat{\text{FI}}_k(x)$ using the squared gradient norm of the log-likelihood with respect to the logits. Here $\pi_k(x)$ always denotes the normalized router weight assigned to head $k$ (after any training-time FI modulation), whereas $\tilde{\pi}_k(x)$ denotes the raw pre-modulation router score used only internally in Eq. (17). We then penalize high-sensitivity allocations via

$$\mathcal{L}_{\text{FI}} = \mathbb{E}_x\left[\sum_{k=1}^{K} \pi_k(x)\, \widehat{\text{FI}}_k(x)\right]. \qquad (13)$$

This Fisher-informed regularizer averages per-head proxies under the mixture weights $\pi_k(x)$, so heads that are both frequently selected and highly sensitive incur a larger penalty. We further add two auxiliary regularizers: an energy-based term that discourages excessively large positive energies, and an uncertainty term that shapes predictive entropy:

$$\mathcal{L}_{\text{EBM}} = \mathbb{E}_x\Big[\text{softplus}\big(\text{clip}(E_\psi(f_\theta(x)), -\tau, \tau)\big)\Big] + \mathcal{L}_{\text{EBM}}^{\text{neg}}, \qquad (14)$$

$$\mathcal{L}_{\text{UNC}} = \beta_{\text{id}}\, \mathbb{E}_x\big[\text{H}(\hat{p}(x))\big] - \beta_{\text{ood}}\, \mathbb{E}_{x_{\text{ood}}}\big[\text{H}(\hat{p}(x_{\text{ood}}))\big], \qquad (15)$$

where $\text{H}(\cdot)$ denotes the (Shannon) entropy of the predictive distribution. The uncertainty loss $\mathcal{L}_{\text{UNC}}$ is contrastive: it encourages low entropy for ID samples (first term) and high entropy for OOD samples (second term, subtracted). We use VOS only for GEM-FI: $\mathcal{L}_{\text{EBM}}^{\text{neg}}$ pushes synthetically generated negative samples toward high-energy regions via a margin-based softplus penalty $\text{softplus}(m - E_{\text{neg}})$ on VOS-synthesized negatives. We treat VOS as an auxiliary boundary-sharpening mechanism within the full GEM-FI pipeline rather than as the core architectural contribution. The same clipping threshold $\tau$ is reused in (14) to prevent energy values from reaching numerically unstable magnitudes. Baselines are trained following their standard protocols without VOS. Putting these components together, the overall training objective for GEM-FI is:

$$\mathcal{L}_{\text{GEM-FI}} = \mathcal{L}_{\text{mix}} + \lambda_{\text{FI}}\mathcal{L}_{\text{FI}} + \lambda_{\text{EBM}}\mathcal{L}_{\text{EBM}} + \lambda_{\text{UNC}}\mathcal{L}_{\text{UNC}}. \qquad (16)$$

Beyond this loss-level regularization, we also use the Fisher proxy to modulate mixture weights during training. Specifically, we compute a per-head proxy $\widehat{\text{FI}}_k(x)$ as the squared $L_2$ norm of the gradient of the log-likelihood with respect to the logits. To ensure bounded and stable modulation, we normalize the proxies across heads to obtain relative sensitivity scores in $[0, 1]$. Let $\tilde{\pi}(x)$ denote the raw softmax output of the router $h_\omega$, and define $\bar{\text{FI}}_k(x) = \widehat{\text{FI}}_k(x) / \big(\sum_j \widehat{\text{FI}}_j(x) + \epsilon\big)$. During training, we reweight the mixture scores as

$$\tilde{\pi}_k^{\text{mod}}(x) \propto \tilde{\pi}_k(x)\, \exp\big(\lambda_{\text{FI}}(1 - \bar{\text{FI}}_k(x))\big), \qquad (17)$$

and renormalize to the simplex using a small smoothing constant for numerical stability: $\pi_k(x) = \big(\tilde{\pi}_k^{\text{mod}}(x) + \epsilon'\big) / \sum_j \big(\tilde{\pi}_j^{\text{mod}}(x) + \epsilon'\big)$, where $\epsilon'$ is set to a small value in all experiments (e.g., $10^{-4}$).

This FI-aware modulation upweights heads with *lower* Fisher sensitivity (i.e., more stable predictions) and is applied *only during training*; at inference, mixture weights are computed directly from the router without FI modulation. Empirically, this design stabilizes mixture allocations and reduces head dominance on challenging OOD examples. Intuitively, the regularizer $\sum_k \pi_k(x)\widehat{\text{FI}}_k(x)$ discourages allocating high weight to locally sensitive heads, while (17) enforces the same preference directly at the mixture level during training.

For implementation details and pseudocode aligned with our training pipeline, see Appendix D.

## 3. Theoretical Insights

We sketch why the proposed components—the bounded, learnable gate in GEM-CORE and the FI-aware mixture in GEM-MIX/GEM-FI—can smooth confidence, encourage distance-informed behavior, and stabilize mixture allocations.

### 3.1. Confidence smoothing via bounded energy-to-gate mapping

Intuitively, if the backbone, energy head, and integration gate are smooth and the final gate is bounded away from 0 and 1, then the evidential outputs inherit this smoothness: nearby inputs cannot induce arbitrarily large changes in evidence or predictive confidence.

**Assumption 3.1** (Lipschitz components). Assume:

- The backbone $f_\theta$ is $L_f$-Lipschitz: $\|f_\theta(x) - f_\theta(x')\| \leq L_f\|x - x'\|$.

- The classifier head $g_\phi$ is $L_g$-Lipschitz.

- The energy head $E_\psi$ is $L_E$-Lipschitz.

- The integration gate $G_\eta([z, \hat{s}])$ is $L_G$-Lipschitz in $(z, \hat{s})$ and outputs gates in $[s_{\min}, s_{\max}]$ with $0 < s_{\min} < s_{\max} < 1$.

- The mixture router $h_\omega$ and density scaler $\rho(z)$ are Lipschitz continuous.

In practice, these smoothness assumptions are supported by the use of spectral normalization throughout the main convolutional and linear mappings in the backbone and the auxiliary gating pathway. Since spectral normalization controls the operator norm of each layer, the growth of the composed mapping is correspondingly constrained, making Assumption 3.1 a practically motivated approximation rather than a purely abstract idealization. We do not claim that spectral normalization alone proves all global smoothness properties of the full network, but it provides an explicit architectural mechanism that supports the intended bounded-growth behavior used in this analysis.

Since $\hat{s}(x) = \sigma(E_\psi(z))$ is a smooth bounded mapping and $\sigma$ is 1-Lipschitz, $\hat{s}$ is $L_E$-Lipschitz by composition. Combining this with $G_\eta$ yields an $L_s$-Lipschitz gate $s(x)$ for some finite $L_s$.

**Proposition 3.2** (Smoothness of probability-level gating). *Under Assumption 3.1, suppose $p_{\mathrm{mix}}(x) \in \Delta^{C-1}$ is locally Lipschitz in $x$ and the per-class gate satisfies $s(x) \in [s_{\min}, s_{\max}]^C$ with $0 < s_{\min} \leq s_{\max} \leq 1$. Then the probability-level gated prediction $\hat{p}(x) = \mathrm{Normalize}\big(p_{\mathrm{mix}}(x) \odot s(x)\big)$ is locally Lipschitz. In particular, there exists $L_{\hat{p}} > 0$ such that for sufficiently close $x, x'$,*

$$\|\hat{p}(x) - \hat{p}(x')\| \leq L_{\hat{p}} \|x - x'\|. \tag{18}$$

*Sketch.* Both $p_{\mathrm{mix}}(x)$ and $s(x)$ are locally Lipschitz by assumption and construction, hence their elementwise product is locally Lipschitz. The normalization map $v \mapsto v/(\mathbf{1}^\top v)$ is smooth wherever $\mathbf{1}^\top v$ is bounded away from zero. Here, for $v(x) = p_{\mathrm{mix}}(x) \odot s(x)$ we have $\mathbf{1}^\top v(x) = \sum_c p_{\mathrm{mix},c}(x) s_c(x) \geq s_{\min} \sum_c p_{\mathrm{mix},c}(x) = s_{\min}$, so the denominator is uniformly positive in a neighborhood. Therefore, $\hat{p}(x)$ is locally Lipschitz as a composition of locally Lipschitz maps with a smooth normalization.

### 3.2. Distance-informed monotonicity (qualitative calibration)

Beyond local smoothness, we would like confidence to decay as we move away from high-support regions in representation space. Our design couples evidence to a learned energy, which can serve as a lightweight control signal correlated with representation-level support. While we do not impose hard monotonicity constraints or assume an explicit density model, the following idealized picture clarifies the role of the gate. Additional support-conditioned diagnostics are provided in Appendix G.5 (Figs. 16–19).

**Assumption 3.3** (Energy–support alignment (empirical)). There exists a representation-level support surrogate $\rho(z)$ such that, on average, higher energy $E_\psi(z)$ is associated with lower support $\rho(z)$ (empirical anti-correlation). We treat $\rho(z)$ as a generic support indicator; the analysis requires only that it is monotonically related to feature support and does not rely on $\rho$ being a calibrated density

model. In our implementation, we use a GMM-based estimator $\rho(z) = \sigma(\log p_{\mathrm{GMM}}(z))^\gamma$, where $p_{\mathrm{GMM}}$ is fit to ID training features. We introduce an *energy pre-gate* $\hat{s}_E(x) = \sigma(E_\psi(z))$, which is monotonically increasing in energy, as an intermediate summary of the energy signal. The integration network $G_\eta$ maps $[z, \hat{s}_E(x)]$ to per-class gates $s(x) \in [s_{\min}, s_{\max}]^C$. We do *not* enforce a hard monotonic relationship between $E_\psi$ and the final gates; instead, the model learns end-to-end to suppress evidence in lower-support regions. This behavior is a learned outcome of training, not an architectural constraint.

In a simplified single-head setting with logits $u(x)$ and gate $s(x)$, define the top-class margin $m(z) = u_y(z) - \max_{c \neq y} u_c(z)$.

**Proposition 3.4** (Monotone suppression away from the support (idealized)). *Under Assumption 3.3, as we move away from the support, (i) the total evidence $\alpha_0(x)$ weakly decreases due to decay of the density scaler $\rho(z)$, and (ii) if the logit margin $m(z)$ does not increase fast enough to compensate for the shrinking gate $s(x)$, the top-class confidence $p_y(x)$ also weakly decreases.*

*Sketch.* The total evidence is $\alpha_0(x) = \sum_c (\rho(z) \exp(\tilde{u}_c) + \epsilon) \approx \rho(z) \sum_c \exp(\tilde{u}_c)$. Under Assumption 3.3, as we move away from the support, the density proxy $\rho(z)$ decays toward zero. If the logit terms $\exp(\tilde{u}_c)$ do not grow exponentially faster than $\rho(z)$ decays, then $\rho(z) \exp(\tilde{u}_c)$ vanishes, leading to $\alpha_0(x) \to C\epsilon$ (minimal evidence). Thus, total evidence decreases in low-support regions.

We emphasize that Proposition 3.4 provides only a conditional guarantee under the stated assumptions; the empirical energy–support alignment should therefore be interpreted as a consistent empirical regularity rather than an architectural invariant.

Full details of the experimental setup are provided in Appendix E.

## 4. Experiments

We evaluate the GEM family on classification, calibration, OOD detection, and robustness under distribution shift. Our experiments are designed to answer the following questions:

*Q1.* How do GEM models compare to EDL and DAEDL in terms of OOD detection?

*Q2.* Do GEM-MIX and GEM-FI preserve or improve ID accuracy and confidence calibration?

*Q3.* What is the contribution of the energy gate, mixture-of-beliefs, and FI regularization?

We first evaluate OOD detection performance (Q1) and report AUPR scores for aleatoric and epistemic uncertainty across common ID→OOD shifts (Table 1). Next, we assess ID accuracy, confidence calibration, and misclassification

detection (Q2) on CIFAR-10 (Table 2). Finally, to verify the necessity of each design choice, we provide ablations over gating, mixture, and FI stabilization (Q3; Table 3).

Unless otherwise stated, all results are averaged over five random seeds (0, 1, 2, 3, 42) and reported as mean ± standard deviation.

### 4.1. OOD Detection

To address Q1, we evaluate OOD detection on four benchmark pairs: MNIST→KMNIST, MNIST→FMNIST, CIFAR-10→SVHN, and CIFAR-10→CIFAR-100. These cover digit-domain grayscale shifts and natural-image shifts, from cross-domain (CIFAR-10→SVHN) to fine-grained, near-OOD settings (CIFAR-10→CIFAR-100).

Table 1 summarizes OOD detection AUPR under several scoring rules, including a simple aleatoric score (maximum softmax probability (MAXP)) and epistemic-leaning scores for evidential models.

Across digit-domain shifts, all GEM variants achieve near-ceiling AUPR and slightly improve over DAEDL and earlier baselines, with GEM-MIX and GEM-FI yielding the strongest epistemic scores. On natural-image benchmarks, GEM-FI attains the best AUPR on CIFAR-10→SVHN, surpassing both DAEDL and Re-EDL.

CIFAR-10→CIFAR-100 remains the most challenging setting: GEM-FI achieves 90.30% aleatoric AUPR, outperforming DAEDL (88.16%) and Re-EDL (87.57%). This near-OOD shift shares low-level statistics and semantic structure with ID, so density cues can remain high and uncertainty separation is harder than in far-OOD shifts (e.g., SVHN), which have distinct textures and color distributions. Thus, GEM performs well in both far-OOD detection and challenging near-OOD scenarios.

Additional results, including AUROC, are reported in Appendix F.1. We also report AUPR based on total evidence $\alpha_0$ and mixture-aware proxies such as MI, with energy and predictive entropy as reference metrics. Results for CIFAR-10→TinyImageNet are provided in Appendix F.11 (Table 12).

**Precision–Recall and ROC curves.** Figure 4 compares PR and ROC curves for OOD detection (CIFAR-10 as ID, SVHN as OOD). In the PR view, GEM variants achieve markedly higher precision at moderate-to-high recall than evidential baselines, indicating fewer false alarms at higher coverage: EDL reaches AUPR 78.87, while GEM-CORE and GEM-MIX improve to 93.87 and 93.72, respectively (GEM-FI: 92.59). The ROC curves show a similar trend, with GEM dominating in the low-FPR regime: AUROC increases from 81.06 (EDL) and 89.24 (DAEDL) to 93.65 (GEM-CORE/GEM-FI) and 93.08 (GEM-MIX). These gains are consistent with the separation induced by learned gating (and mixture heads), enabling high-precision detection while preserving single-pass inference compared to multi-pass ensembling. Extended versions of Figure 4 are

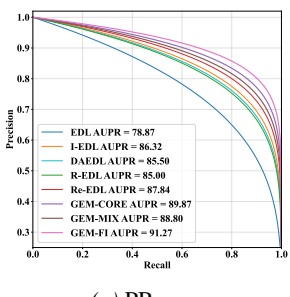 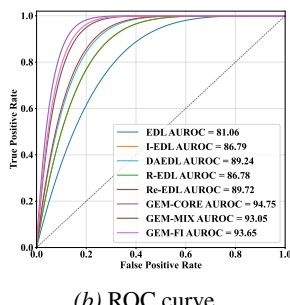

*(a) PR curve*       *(b) ROC curve*

*Figure 4.* PR and ROC curves for OOD detection on CIFAR-10 (ID) vs. SVHN (OOD).

provided in Appendix F.2 and F.3.

As an additional stress test, we evaluate GEM under common distribution-shift and corruption benchmarks (Appendix F.4).

### 4.2. Image Classification and Confidence Calibration

To address Q2, we report ID test accuracy, misclassification-detection AUPR, and the Brier score on CIFAR-10. Table 2 summarizes these metrics for posterior-network and evidential baselines, along with our GEM-based models. Among prior methods, DAEDL provides the strongest accuracy–calibration trade-off. Compared to DAEDL, GEM-FI improves test accuracy from 91.11% to 93.75%, reduces the Brier score from 14.27 to 6.81, and increases misclassification-detection AUPR from 99.08 to 99.93, indicating more accurate predictions with better-calibrated confidence.

For reference, a well-calibrated softmax ResNet-18 on CIFAR-10 (without evidential training) typically yields Brier×100 ≈ 15–20. Our GEM-CORE attains an unusually low Brier × 100 ≈ 1.27 because probability-space gating produces extremely sharp predictions (near-deterministic when correct), which substantially lowers the squared-error term in the Brier score. In contrast, the mixture variants (GEM-MIX, GEM-FI) are less sharp due to mixture averaging, resulting in Brier values closer to typical ranges; this behavior reflects the gating mechanism rather than a calculation issue. Additional comparisons against classical TS are provided in Appendix F.5.

### 4.3. Ablation Study

Finally, Q3 examines the contribution of each GEM-FI component on CIFAR-10 and its standard OOD pairs. Table 3 reports an ablation over seven switches: (i) spectral normalization (SN); (ii) the energy-gated evidential module (CORE); (iii) the mixture of evidential heads (MIX, i.e., GEM-MIX); (iv) FI regularization (FI-Reg); (v) Fisher-based modulation of mixture weights (FI-Mod); (vi) the energy-based module (EBM); and (vii) uncertainty decomposition (UNC). SN alone improves over the baseline, highlighting its stabilizing role for training and calibration. CORE on top of SN yields a further gain, indicating complementary benefits be-

*Table 1.* AUPR scores of OOD detection based on aleatoric and epistemic uncertainty. $A \to B$ denotes that $A$ is used as the ID dataset and $B$ as the OOD dataset.

| Method | Venue | MNIST → KMNIST | | MNIST → FMNIST | | CIFAR-10 → SVHN | | CIFAR-10 → CIFAR-100 | |
|---|---|---|---|---|---|---|---|---|---|
| | | Alea.↑ | Epis.↑ | Alea.↑ | Epis.↑ | Alea.↑ | Epis.↑ | Alea.↑ | Epis.↑ |
| DROPOUT | ICML16 | 94.00 ± 0.10 | – | 96.56 ± 0.20 | – | 51.39 ± 0.10 | – | 45.57 ± 1.00 | – |
| KL-PN | NeurIPS18 | 92.97 ± 1.20 | 93.39 ± 1.00 | 98.14 ± 0.80 | 98.16 ± 0.00 | 43.96 ± 1.90 | 43.23 ± 2.30 | 61.41 ± 2.80 | 61.53 ± 3.40 |
| EDL | NeurIPS18 | 97.02 ± 0.80 | 96.31 ± 2.00 | 98.10 ± 0.40 | 97.84 ± 0.40 | 78.87 ± 3.50 | 79.32 ± 1.70 | 84.30 ± 0.70 | 84.80 ± 1.00 |
| RKL-PN | NeurIPS19 | 60.76 ± 2.90 | 53.76 ± 3.40 | 78.45 ± 3.10 | 72.18 ± 3.60 | 53.61 ± 1.10 | 49.37 ± 0.80 | 55.42 ± 2.60 | 54.74 ± 2.80 |
| POSTNET | NeurIPS20 | 95.75 ± 0.20 | 94.59 ± 0.30 | 97.72 ± 0.20 | 97.24 ± 0.20 | 80.21 ± 0.20 | 77.71 ± 0.40 | 81.96 ± 0.80 | 82.06 ± 0.80 |
| *I*-EDL | ICML23 | 98.34 ± 0.20 | 98.33 ± 0.20 | 98.86 ± 0.30 | 98.86 ± 0.30 | 86.32 ± 2.40 | 85.92 ± 2.30 | 85.55 ± 0.70 | 84.84 ± 0.60 |
| DAEDL | ICML24 | 99.90 ± 0.00 | 99.92 ± 0.00 | 99.83 ± 0.00 | 99.87 ± 0.00 | 85.50 ± 1.40 | 85.54 ± 1.40 | 88.16 ± 0.10 | 88.19 ± 0.10 |
| R-EDL | ICLR24 | – | 98.69 ± 0.20 | – | 99.29 ± 0.12 | 85.00 ± 1.22 | 85.00 ± 1.22 | 87.72 ± 0.31 | 87.73 ± 0.31 |
| CEDL+ | ESWA25 | 99.88 ± 0.07 | 99.89 ± 0.07 | 98.17 ± 0.01 | 98.10 ± 0.01 | 89.30 ± 0.34 | 89.16 ± 0.65 | 79.57 ± 0.57 | 77.16 ± 0.52 |
| LTS | MVA25 | 98.17 ± 0.85 | 99.94 ± 0.03 | 99.65 ± 0.12 | 99.80 ± 0.12 | 78.63 ± 0.96 | 80.64 ± 0.88 | 71.23 ± 0.79 | 85.33 ± 0.65 |
| Re-EDL | TPAMI25 | – | 99.03 ± 0.28 | – | 99.65 ± 0.09 | 87.84 ± 0.96 | 89.89 ± 1.39 | 87.57 ± 0.23 | 88.30 ± 0.16 |
| GEM-CORE | | 99.93 ± 0.01 | 99.90 ± 0.03 | 99.99 ± 0.00 | 99.97 ± 0.01 | 89.87 ± 0.33 | 87.80 ± 0.15 | 89.35 ± 0.23 | 84.00 ± 0.40 |
| GEM-MIX | | 99.94 ± 0.01 | 99.94 ± 0.02 | 99.99 ± 0.00 | 99.96 ± 0.04 | 88.80 ± 0.24 | 90.60 ± 0.23 | 84.98 ± 0.10 | 84.46 ± 0.20 |
| GEM-FI | | **99.95 ± 0.00** | **99.96 ± 0.01** | **99.99 ± 0.00** | **99.99 ± 0.00** | **91.27 ± 0.29** | **92.59 ± 0.31** | **90.30 ± 0.06** | **90.20 ± 0.06** |

*Table 2.* Image classification and confidence calibration on CIFAR-10.

| Method | Test Acc.↑ | AUPR↑ | Brier (×100)↓ |
|---|---|---|---|
| Dropout | 82.84 ± 0.10 | 97.15 ± 0.00 | 27.15 ± 0.20 |
| KL-PN | 27.46 ± 1.70 | 50.61 ± 4.00 | 87.28 ± 1.00 |
| RKL-PN | 64.76 ± 0.30 | 86.11 ± 0.40 | 54.73 ± 0.40 |
| PostNet | 84.85 ± 0.00 | 97.76 ± 0.20 | 22.84 ± 0.00 |
| EDL | 83.55 ± 0.60 | 97.86 ± 0.30 | 23.38 ± 0.20 |
| *I*-EDL | 89.20 ± 0.30 | 98.72 ± 0.10 | 35.20 ± 0.80 |
| DAEDL | 91.11 ± 0.20 | 99.08 ± 0.00 | 14.27 ± 0.20 |
| CEDL+ | 93.07 ± 0.06 | 98.82 ± 0.01 | 15.02 ± 0.03 |
| LTS | 93.13 ± 0.10 | 98.87 ± 0.01 | 14.97 ± 0.03 |
| R-EDL | 90.09 ± 0.31 | 98.98 ± 0.05 | 18.15 ± 0.50 |
| RE-EDL | 90.13 ± 0.21 | 98.81 ± 0.01 | 14.95 ± 0.47 |
| GEM-CORE | 93.34 ± 0.10 | 99.87 ± 0.01 | **1.27 ± 0.02** |
| GEM-MIX | 93.27 ± 0.31 | 99.93 ± 0.02 | 6.97 ± 0.03 |
| GEM-FI | **93.75 ± 0.36** | **99.94 ± 0.01** | 6.81 ± 0.01 |

yond SN. MIX further improves OOD detection, supporting single-pass mixtures for multi-modal epistemic uncertainty. Alone, FI-Reg or FI-Mod can reduce OOD AUPR, suggesting Fisher shaping is most effective with the energy-based mechanism and uncertainty decomposition. This pattern clarifies the intended FI–EBM synergy in our design: the FI-based regularization and modulation terms are not meant to operate as standalone mechanisms, but rather to refine routing once the energy-based components have already shaped a meaningful support signal. Without EBM/UNC, the learned energy landscape can remain comparatively weak or noisy, in which case FI-based routing may amplify unstable head assignments instead of improving epistemic separation. Once the energy-based mechanism provides a clearer notion of support, however, FI modulation becomes substantially more effective at preventing head collapse and stabilizing mixture allocation. With EBM and UNC enabled, the full GEM-FI configuration achieves the strongest overall performance, including improved aleatoric/epistemic separation on CIFAR-10→SVHN (91.27/92.59) and CIFAR-10→CIFAR-100 (90.30/90.20). Removing SN from the full model degrades both accuracy and uncertainty quality, em-

phasizing its stabilizing role during training. Additional controlled comparisons covering fair VOS exposure, replacing VOS with simple feature-space noise, removing the supplementary density scaler $\rho(z)$, and head-diversity diagnostics are reported in Appendix G.6, Appendix G.7, and Appendix G.9.

**Qualitative evidence geometry.** Figure 5 compares how different evidential formulations can yield distinct uncertainty structures even when they produce identical class predictions. GEM-CORE concentrates evidence sharply around a single mode, resulting in high confidence but limited representation of epistemic alternatives. In contrast, GEM-MIX distributes evidence across multiple mixture components, enabling a multi-modal representation of uncertainty. Finally, GEM-FI regularizes mixture allocations using Fisher information, balancing concentration and diversity to yield smoother Dirichlet geometries and more stable uncertainty estimates. Figure 6 visualizes the feature embeddings learned by GEM-FI. Compact clustering of ID classes and clear separation from OOD data highlight the effect of Fisher-informed regularization in shaping the latent space. Additional parameter-sensitivity analyses are provided in Appendix G.

## 5. Conclusion

We presented GEM, a family of single-pass, distance-informed evidential models. GEM-CORE learns an in-model energy-to-gate mapping that directly modulates class probabilities via probability-space gating; GEM-MIX extends this design with a lightweight belief mixture to capture multi-modal epistemic structure; and GEM-FI stabilizes mixture allocations via a FI-informed regularizer and FI-based modulation of mixture weights. Across MNIST/CIFAR-10, common OOD pairs, and corruption suites, GEM-based models consistently improve calibration and strengthen OOD separation relative to a DAEDL-style baseline, while preserving single-pass inference.

*Table 3.* Ablation on CIFAR-10. ✓ indicates an enabled component.

| | | | | | | | CIFAR-10 → SVHN | | | | CIFAR-10 → CIFAR-100 | |
| --- | --- | --- | --- | --- | --- | --- | --- | --- | --- | --- | --- | --- |
| SN | CORE | MIX | FI-Reg | FI-Mod | EBM | UNC | Test Acc.↑ | AUPR↑ | Alea.↑ | Epis.↑ | Alea.↑ | Epis.↑ |
| ✗ | ✗ | ✗ | ✗ | ✗ | ✗ | ✗ | 83.55 ± 0.60 | 97.86 ± 0.20 | 78.87 ± 3.50 | 79.12 ± 3.70 | 84.30 ± 0.70 | 84.18 ± 0.70 |
| ✓ | ✗ | ✗ | ✗ | ✗ | ✗ | ✗ | 91.00 ± 0.40 | 99.20 ± 0.10 | 85.50 ± 1.50 | 85.20 ± 1.60 | 87.00 ± 0.50 | 86.50 ± 0.55 |
| ✓ | ✓ | ✗ | ✗ | ✗ | ✗ | ✗ | 93.34 ± 0.10 | 99.87 ± 0.01 | 89.87 ± 0.33 | 87.80 ± 0.15 | 89.35 ± 0.23 | 84.00 ± 0.40 |
| ✓ | ✓ | ✓ | ✗ | ✗ | ✗ | ✗ | 93.27 ± 0.31 | 99.93 ± 0.02 | 88.80 ± 0.24 | 90.60 ± 0.23 | 84.98 ± 0.10 | 84.46 ± 0.20 |
| ✓ | ✓ | ✓ | ✓ | ✗ | ✗ | ✗ | 93.40 ± 0.15 | 89.68 ± 0.30 | 93.09 ± 0.40 | 87.78 ± 0.50 | 85.01 ± 0.35 | 80.14 ± 0.45 |
| ✓ | ✓ | ✓ | ✗ | ✓ | ✗ | ✗ | 93.50 ± 0.12 | 87.35 ± 0.25 | 90.60 ± 0.35 | 75.38 ± 0.55 | 84.25 ± 0.40 | 71.77 ± 0.50 |
| ✓ | ✓ | ✓ | ✓ | ✓ | ✗ | ✗ | 93.60 ± 0.10 | 84.42 ± 0.20 | 90.50 ± 0.30 | 75.01 ± 0.45 | 84.11 ± 0.30 | 72.03 ± 0.40 |
| ✓ | ✓ | ✓ | ✓ | ✓ | ✓ | ✗ | 93.70 ± 0.08 | 91.16 ± 0.15 | **93.95 ± 0.35** | 94.93 ± 0.40 | 86.26 ± 0.25 | 87.37 ± 0.35 |
| ✓ | ✓ | ✓ | ✓ | ✓ | ✓ | ✓ | **93.75 ± 0.36** | **99.93 ± 0.01** | 91.27 ± 0.29 | 92.59 ± 0.31 | **90.30 ± 0.08** | **90.20 ± 0.07** |
| ✗ | ✓ | ✓ | ✓ | ✓ | ✓ | ✓ | 92.10 ± 0.25 | 85.92 ± 0.20 | 92.23 ± 0.40 | 78.64 ± 0.50 | 83.80 ± 0.35 | 72.44 ± 0.45 |

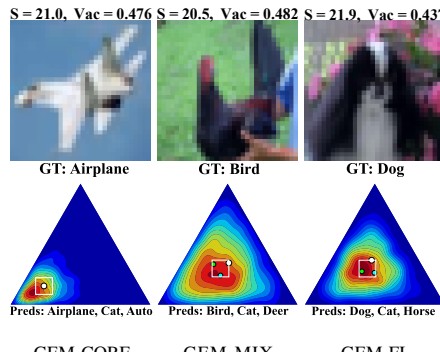

*Figure 5.* Comparison of three GEM variants on CIFAR-10 test images. The input is shown at the top of each column, and the induced Dirichlet distribution is visualized on the probability simplex below. Annotations $S$ (total evidence) and $V_{ac}$ (vacuity) summarize the resulting uncertainty geometry.

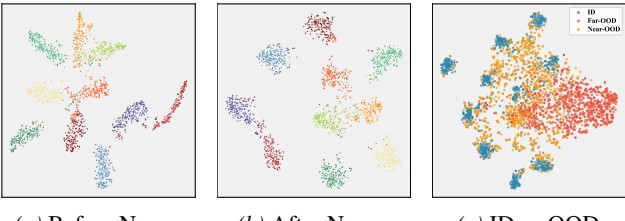

*(a)* Before Norm.    *(b)* After Norm.    *(c)* ID vs OOD

*Figure 6.* t-SNE visualization of feature embeddings for GEM-FI.

**Limitations and future work.** GEM uses learned energy as an internal control signal for evidence gating, not as a calibrated estimator of representation-level support. Accordingly, alignment between energy and any support proxy (e.g., $k$NN distance) is empirical and may vary across regimes, architectures, and datasets; we provide no monotonicity guarantee. This limits interpreting energy as a direct support measure and suggests that stronger guarantees may require explicit support modeling or additional regularization. Near-OOD shifts with high semantic/visual overlap are intrinsically harder and often yield weaker uncertainty separability, leaving less headroom for any method. Finally, mixture routing introduces hyperparameter sensitivity and modest compute overhead, motivating more robust routing and tighter calibration guarantees (see Appendix H). Our empirical evidence is currently concentrated on standard MNIST/CIFAR-style benchmarks and associated corruption/OOD protocols, so our claims are intentionally scoped to those settings rather than to ImageNet-scale evaluation.

## Impact Statement

This paper studies uncertainty estimation for deep classification via single-pass evidential models. Improved calibration and more reliable OOD detection can positively impact safety-critical deployments by helping systems abstain or defer when inputs are unsupported by the training data, thereby reducing overconfident failures. Potential risks include inappropriate over-reliance on uncertainty scores as a substitute for domain-specific validation, as well as misuse in high-stakes settings (e.g., surveillance or automated decision-making) where errors or dataset biases can cause harm. To mitigate these risks, we recommend reporting calibration and OOD metrics under multiple shifts, auditing performance across relevant subpopulations, and communicating uncertainty as one component in a broader human-in-the-loop decision process.

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

## A. Code and Reproducibility

Code and reproduction instructions are available at: `https://github.com/Marcorazhan/GEM-FI`.

## B. Notation Summary

Table 4 collects the main symbols used in Sections 2.1–2.3 and fixes the distinction between single-head, per-head, gating, and routing quantities.

## C. Related Work

### C.1. Single-Pass Evidential Models and Density-Aware Extensions

EDL provides single-pass predictive uncertainty by parameterizing a Dirichlet distribution and interpreting its concentration as evidence (Sensoy et al., 2018). Large-scale evaluations caution that modern networks can remain miscalibrated under distribution shift (Ovadia et al., 2019). To encode data support more explicitly, Prior Networks (Malinin & Gales, 2019) shape Dirichlet targets with priors, and Posterior Networks (Charpentier et al., 2020) parameterize target Dirichlet distributions. Density-aware variants rescale evidential outputs using feature-space likelihoods; DAEDL employs an offline Gaussian surrogate such as GDA (Murphy, 2012; Bishop, 2006), which improves calibration under shift but leaves the density term decoupled from end-to-end learning. Mixture-style evidential models (Ryu et al., 2024) capture ambiguity via multiple Dirichlet components with learned mixing. Beyond architecture, Dirichlet calibration (Kim et al., 2024) for modern networks has been revisited, semi-supervised signals (Cheng et al., 2024) have been used to improve shift-aware confidence, and gated evidential formulations (Zhang et al., 2024) report calibration gains by explicitly modulating evidential outputs. FI-informed evidential training has also been explored (Deng et al., 2023). Table 5 summarizes how GEM-FI differs from closely related single-pass evidential and density-aware methods along key architectural and algorithmic dimensions. GEM-FI integrates in-model support gating, multi-modal epistemic mixtures, and Fisher-inspired stabilization for routing in a unified single-pass evidential framework.

### C.2. OOD Baselines and Post hoc Energy Methods

Non-energy baselines remain standard references for OOD detection: MAXP (MaxP) (Hendrycks & Gimpel, 2017), ODIN (Liang et al., 2017) with input perturbations and TS, and Mahalanobis scoring (Lee et al., 2018) in feature space. Energy-based views (Grathwohl et al., 2020; Liu et al., 2020) reinterpret discriminative classifiers as implicit energy models and have reported improved separability in some settings between ID and OOD examples than softmax confidence. Follow-ups analyze theoretical conditions for energy-based separability (Morteza & Li, 2022), explore architectural variants such as masked energy models (He et al., 2023), and study calibration and representation-shift effects for energy scores (Zhou et al., 2024; Huang et al., 2024). Unified, post-hoc calibration frameworks under distribution shift have also been proposed (Sehwag et al., 2024; Romero et al., 2024). For broader overviews of OOD detection, see the survey by Yang et al. (2024).

## D. Implementation-Aligned Pseudocode and Notes

**Implementation vs. theory.** Our implementation follows the GEM-FI design in Figure 3b but makes two choices that we state explicitly to avoid ambiguity. First, the learned integration gate is applied *after* mixture aggregation, i.e., we form a mixture predictive mean and then apply a per-class gate in probability space, followed by renormalization. Second, each evidential head parameterizes Dirichlet concentrations directly as $\alpha_k = \exp(\text{clip}(u_k)) + \varepsilon$ (no explicit "+1" offset), matching the code path used for all GEM-FI results. The Fisher-inspired quantity used by GEM-FI is a tractable sensitivity proxy computed from per-sample gradients of the component log-probability with respect to the component logits; it is used both to (i) modulate mixture weights during the forward pass and (ii) regularize training via an additional loss term. Finally, the implementation multiplies each component concentration by a per-sample feature-density score before forming expectations, which sharpens or suppresses evidence depending on feature support.

**Virtual Outlier Synthesis (VOS).** For GEM-FI, we employ VOS to generate synthetic OOD samples near the decision boundary. We sample $\epsilon \sim \mathcal{N}(0, 1)$ and generate virtual outliers $v_k$ by sampling from the class-conditional Gaussian estimates in the feature space. We train with a VOS regularization weight of 0.1, a warmup of 10 epochs, and synthesize outcomes to enforce low evidence on these virtual points. This auxiliary loss is used only in the full GEM-FI configuration as a boundary-sharpening mechanism to improve OOD separability; it is not the primary source of the core gating or mixture behavior.

**Energy signal and robust calibration.** We compute a learned energy head $E_\psi(z)$ and, for reference, a density-based GMM energy $E_{\text{gmm}}(z) = -\log \sum_k \exp(\log p(z \mid k))$. For evaluation-time scaling we select the energy source with the larger robust dynamic range (1–99% quantile span), which is typically the learned energy head in our runs. For GEM-FI with VOS-EBM enabled, we disable the final $\tanh$ on the energy head; when VOS is not used, enabling $\tanh$ can help prevent sigmoid-gate saturation. When an energy-to-confidence scalar is needed (e.g., for reporting an energy-based shift score), we use $s = \text{clip}\left(1 - \frac{E - E_{\min}}{E_{\max} - E_{\min}}, 0, 1\right)$ with $(E_{\min}, E_{\max})$ taken from 1–99% quantiles; if the range is numerically tight, we fall back to a logits-based energy $-\log \sum_c \exp(u_c)$.

*Table 4.* Notation summary for the main method sections. Single-head quantities are used by EDL/GEM-CORE; superscript $(k)$ denotes the corresponding quantity for mixture head $k$ in GEM-MIX/GEM-FI.

| Symbol | Meaning |
|---|---|
| $x, y, C$ | input, class label, and number of classes |
| $f_\theta(x) = z \in \mathbb{R}^d$ | shared backbone and resulting feature embedding |
| $u_c(x),\ \alpha_c(x),\ p_c(x)$ | single-head logit, Dirichlet concentration, and predictive mean for class $c$ |
| $u_c^{(k)}(x),\ \alpha_c^{(k)}(x),\ p_c^{(k)}(x)$ | per-head counterparts for mixture component $k$ |
| $\alpha_0(x),\ \alpha_0^{(k)}(x)$ | total concentration (single-head / head $k$) |
| $E_\psi(z),\ E(x)$ | learned energy head and resulting scalar energy |
| $\hat{s}(x)$ | intermediate scalar gate signal obtained from energy |
| $s_c(x)$ | final class-wise integration gate for class $c$ |
| $\rho(z)$ | auxiliary feature-density scaling term applied to concentrations |
| $\pi_k(x)$ | normalized router weight for head $k$ |
| $\tilde{\pi}_k(x)$ | raw pre-modulation router score before FI-based reweighting |
| $p_{\text{mix}}(x)$ | mixture predictive mean before class-wise probability gating |
| $\hat{p}(x)$ | final gated and renormalized predictive distribution |
| $\widehat{\text{FI}}_k(x)$ | Fisher-sensitivity proxy for head $k$ |

*Table 5.* Comparison of GEM-FI with closely related single-pass evidential and density-aware methods.

| Method | End-to-end density | Single-pass | Multi-modal epistemic | In-model gating | FI for routing |
|---|---|---|---|---|---|
| DAEDL (Yoon & Kim, 2024) | ✗ | ✓ | ✗ | ✗ | ✗ |
| Ryu et al. (Ryu et al., 2024) | ✓ | ✓ | ✓ | ✗ | ✗ |
| Deng et al. (Deng et al., 2023) | ✓ | ✓ | ✗ | ✗ | ✓ |
| Zhang et al. (Zhang et al., 2024) | ✓ | ✓ | ✗ | ✓ | ✗ |
| **GEM-FI** | ✓ | ✓ | ✓ | ✓ | ✓ |

**Implementation alignment with DAEDL.** While the canonical EDL formulation often uses $\alpha = e + \mathbf{1}$ to encode an explicit Dirichlet(1) base concentration, our implementation follows DAEDL and parameterizes $\alpha$ directly via exponentiated (clipped) logits. Concretely, we use $\alpha = \exp(\tilde{u}) + \epsilon$ with $\epsilon = 10^{-8}$. This ensures $\alpha > 0$ and stable training while matching the DAEDL-style evidential parameterization. Accordingly, all uncertainty quantities that depend on $\alpha$ (e.g., $\alpha_0 = \sum_c \alpha_c$ and vacuity-like measures) are computed using (2) without adding an extra $+1$.

**Training objective (implementation-aligned).** Given $\hat{p}$ from Algorithm 1, the predictive loss is $\mathcal{L}_{\text{pred}} = -\log \hat{p}_y$. We regularize each component with a Dirichlet prior via a mixture-weighted KL term $\mathcal{L}_{\text{KL}} = \sum_{k=1}^{K} \mathbb{E}\big[\pi_k\, \text{KL}(\text{Dir}(\bar{\alpha}_k) \,\|\, \text{Dir}(\mathbf{1}))\big]$. When Fisher modulation is enabled, we add $\mathcal{L}_{\text{FI}} = \mathbb{E}\big[\sum_{k=1}^{K} \pi_k\, FI_k\big]$ and an additional expected-trace penalty $\beta\, \mathbb{E}[FI]$ as implemented. The total loss is $\mathcal{L} = \mathcal{L}_{\text{pred}} + \lambda_{\text{KL}}\mathcal{L}_{\text{KL}} + \lambda_{\text{FI}}\mathcal{L}_{\text{FI}} + \beta\, \mathbb{E}[FI]$.

# E. Experimental Setup

**Datasets.** We evaluate ID classification on MNIST (LeCun et al., 1998) and CIFAR-10 (Krizhevsky, 2009). MNIST contains 60,000 training and 10,000 test grayscale images of size $28 \times 28$, and CIFAR-10 consists of 50,000 training and 10,000 test RGB images of size $32 \times 32$. For OOD evaluation, we use FashionMNIST (Xiao et al., 2017) and KMNIST (Clanuwat et al., 2018) as OOD datasets for MNIST, and SVHN (Netzer et al., 2011) and CIFAR-

100 (Krizhevsky, 2009) as OOD datasets for CIFAR-10. To assess robustness under distributional shift, we additionally evaluate on MNIST-C (Mu & Gilmer, 2019) and CIFAR-10-C (Hendrycks & Dietterich, 2019). MNIST-C consists of 15 corruption types with a fixed (tuned) severity for each corruption, while CIFAR-10-C applies 19 corruption types to the test set across 5 severity levels.

**Backbones and models.** For MNIST we use a small CNN; for CIFAR-10 we use a ResNet-18 backbone. Spectral normalization is applied to all convolutional and linear layers. Our main variants are:

- GEM-CORE: a single evidential head with learned energy $E_\psi(z)$ and bounded gate $s(x)$ that modulates Dirichlet evidence (Sec. 2.1).

- GEM-MIX: a mixture of $K$ evidential heads with learned mixture weights $\pi(x)$ on shared features (Sec. 2.2).

- GEM-FI: the full model with FI-informed regularization to stabilize mixture allocations (Sec. 2.3).

We compare against DROPOUT, MAXP (Hendrycks & Gimpel, 2017), KL-PN and EDL (Sensoy et al., 2018), ODIN (Liang et al., 2017), Mahalanobis scoring (Lee et al., 2018), RKL-PN and POSTNET (Charpentier et al., 2020), $\mathcal{I}$-EDL (Deng et al., 2023), density-aware DAEDL (Yoon & Kim, 2024), R-EDL (Chen et al., 2024), the recent CEDL+ and LTS models, and finally Re-EDL (Chen et al., 2025).

**DAEDL vs. GEM-FI Dirichlet parameterization.** Table 6 summarizes how the DAEDL baseline and our GEM-FI model instantiate the Dirichlet concentration $\alpha_c^{(k)}$ and the corresponding final predictive mean $\hat{p}_c$ during training. DAEDL uses a single-head softmax with $\alpha_c = \exp(\lambda(x)u_c)$ so that the predictive mean coincides with the standard softmax predictor. (Note: This matches our implementation form where density scaling is applied directly to the

**Algorithm 1** GEM-FI forward pass and training losses (implementation-aligned).

**Require:** minibatch $(x, y)$; backbone $f_\theta$; features $z = f_\theta(x)$; $K$ Dirichlet heads $h_k$; mixture router $g_\phi$; energy network $e_\psi$; integration gate $q_\omega$; temperature $T$; density scaler $d(\cdot)$; Fisher-modulation strength $\lambda_{\mathrm{FI}}$; KL strength $\lambda_{\mathrm{KL}}$.

1: $z \leftarrow f_\theta(x)$
2: $E \leftarrow e_\psi(z)$; $s \leftarrow \sigma(E)$ {scalar gate signal}
3: **for** $k = 1, \ldots, K$ **do**
4:     $u_k \leftarrow h_k(z)/T$
5:     $\alpha_k \leftarrow \exp(\mathrm{clip}(u_k, -10, 10)) + \varepsilon$
6: **end for**
7: $\tilde{\pi} \leftarrow g_\phi([z; s])$ {$K$-way softmax}
8: **if** training and Fisher modulation enabled and gradients enabled **then**
9:     **if** $y$ is not provided **then**
10:        $\hat{y} \leftarrow \arg\max_c \frac{1}{K} \sum_{k=1}^K u_{k,c}$; $y \leftarrow \hat{y}$
11:     **end if**
12:     **for** $k = 1, \ldots, K$ **do**
13:        $FI_k \leftarrow \sum_c \left(\nabla_{u_{k,c}} \log p_k(y \mid x)\right)^2$ {logit-sensitivity proxy via autograd}
14:     **end for**
15:     $\bar{FI} \leftarrow \mathrm{Normalize}(FI)$ across components
16:     $\pi \leftarrow \mathrm{Normalize}\big(\tilde{\pi} \odot \exp(\lambda_{\mathrm{FI}}(1 - \bar{FI}))\big)$
17: **else**{Inference: no gradient computation, no Fisher modulation}
18:     $\pi \leftarrow \tilde{\pi}$ {mixture weights directly from router}
19: **end if**
20: $\rho \leftarrow d(z)$ {per-sample density score}
21: **for** $k = 1, \ldots, K$ **do**
22:     $\bar{\alpha}_k \leftarrow \rho \cdot \alpha_k + \varepsilon$
23: **end for**
24: $p_{\mathrm{mix}} \leftarrow \sum_{k=1}^K \pi_k \cdot \mathbb{E}[\mathrm{Dir}(\bar{\alpha}_k)]$ {$\mathbb{E}[\mathrm{Dir}(\alpha)] = \alpha/\alpha_0$}
25: $g \leftarrow q_\omega(s, z)$ {per-class gates}
26: $\hat{p} \leftarrow \mathrm{Normalize}(p_{\mathrm{mix}} \odot g)$
27: **return** $\hat{p}$ (and optional diagnostics: $E, g, \pi, \{\bar{\alpha}_k\}, FI, \alpha_0$)

*Table 6.* Comparison of the per-component Dirichlet concentration $\alpha_c^{(k)}$ and final predictive mean $\hat{p}_c$ between the DAEDL baseline and the proposed GEM-FI method. Here $u_c$ and $u_c^{(k)}$ denote the logits of a single head and mixture component $k$, respectively, and $w_k$ are the learned mixing weights.

| | DAEDL | GEM-FI |
|---|---|---|
| $\alpha_c^{(k)}$ | $\exp(\lambda(x)u_c)$ | $\exp\left(\mathrm{clip}(u_c^{(k)}, -\tau, \tau)\right) + \varepsilon$ |
| Predictive mean $\hat{p}_c$ | $\dfrac{\exp(\lambda(x)u_c)}{\sum_{c'=1}^C \exp(\lambda(x)u_{c'})}$ | $\sum_{k=1}^K w_k \cdot \dfrac{\alpha_c^{(k)}}{\sum_{c'} \alpha_{c'}^{(k)}}$ |
| FI regularization | Not used | $\lambda_{\mathrm{FI}} \mathcal{L}_{\mathrm{FI}}$ |

log-potential before exponentiation). In contrast, GEM-FI maintains $K$ separate Dirichlet heads, each with concentration $\alpha_c^{(k)} = \exp(\mathrm{clip}(u_c^{(k)}, -\tau, \tau)) + \varepsilon$. The final

predictive mean is computed as a *weighted average of per-component expectations*: $\hat{p}_c = \sum_k w_k \cdot \frac{\alpha_c^{(k)}}{\alpha_0^{(k)}}$. This mixture-of-expectations form (rather than a sum of weighted concentrations) matches our implementation. The last row highlights the additional FI regularization term $\lambda_{\mathrm{FI}} \mathcal{L}_{\mathrm{FI}}$ specific to GEM-FI, which modulates mixture allocations without changing the predictive mean.

**Training protocol.** All models are trained with AdamW, cosine learning-rate decay, and gradient clipping with norm 1.0. Batch size and other key hyperparameters are reported in Table 7. The evidential objective consists of a regression-to-target term plus a KL penalty to the uniform Dirichlet, with optional mixture and FI terms for GEM-MIX and GEM-FI. For the gated models, we clamp the gate to $(s_{\min}, s_{\max}) = (0.1, 0.9)$; when the optional tanh nonlinearity is enabled on the energy head, eval-time desaturation (scaling by 0.5) can be applied. The gated evidential core uses an energy head $E_\psi$ (MLP on $z$; tanh disabled by default) followed by a scalar sigmoid $\hat{s}(x)$ and a small integration gate $G_\eta$ that takes $[z, \hat{s}(x)]$ to produce per-class gates $s_c(x) \in [0.1, 0.9]$, which then scale the predictive distribution in probability space and are renormalized (5). For the mixture variant, $K$ heads are used with mixture weights $\pi = \mathrm{softmax}(h_\omega([z, \hat{s}(x)]))$; this is enabled via -use_mob and sized by -num_components$= K$. All mixture calibration and MI metrics use the mixture predictive. FI regularization is enabled via -use_fi_regularization; the weight -fi_lambda sets $\lambda_{\mathrm{FI}}$ for the loss term and also controls the strength of FI-based modulation of mixture weights. The FI proxy is bounded and computed per head, and an optional small penalty is added on the mean FI across heads. Specifically, the per-head FI proxy is computed as $\widehat{\mathrm{FI}}_k(x) = \|\nabla_{u_k} \log p_k(y \mid x)\|_2^2$, and the normalized version used for modulation is:

$$\bar{\mathrm{FI}}_k(x) = \frac{\widehat{\mathrm{FI}}_k(x)}{\sum_j \widehat{\mathrm{FI}}_j(x) + \epsilon}, \qquad (19)$$

where $u_k = h_k(z)/T$ are the component logits (Algorithm 1), and $p_k(y \mid x)$ is the per-head predictive mean. In the absence of ground truth labels during training (e.g., semi-supervised settings or specific loss evaluations), we use the model's predicted pseudo-label $\hat{y} = \arg\max \sum_k u_k$ to compute the gradient proxy.

Similarly, the uncertainty loss $\mathcal{L}_{\mathrm{UNC}}$ (15) incorporates a contrastive OOD term that encourages low entropy for in-distribution samples and high entropy for OOD samples.

**Metrics.** On ID test sets we report accuracy, NLL, Brier score (defined as $\frac{1}{N} \sum_{i=1}^N \sum_{c=1}^C (\hat{p}_{ic} - y_{ic})^2$, reported as $\times 100$ for readability; note that our implementation uses the standard multi-class Brier score without an additional $1/C$ normalization factor), and ECE with 15 equal-width bins. For OOD and distribution-shift detection we report AUROC and AUPR (positive = OOD or corrupted), using

*Table 7.* Hyperparameters. $\lambda_{\mathrm{FI}}$ used only for GEM-FI. *Dropout in the backbone/classifier head; GEM's internal components (energy network, integration gate) use lower dropout (0.01–0.03) for stability.

| Parameter | MNIST | CIFAR-10 |
|---|---|---|
| Learning rate | $5 \times 10^{-4}$ | $10^{-3}$ |
| Batch size | 64 | 128 |
| Epochs | 50 | 100 |
| $\lambda_{\mathrm{KL}}$ | $10^{-3}$ | $10^{-4}$ |
| $\lambda_{\mathrm{FI}}$ | 0.3 | 0.1 |
| $K$ (heads) | 3 | 3 |
| Dropout* | 0.05 | 0.1 |
| $\beta_{\mathrm{id}}$ | 0.1 | 0.1 |
| $\beta_{\mathrm{ood}}$ | 0.1 | 0.1 |
| Weight decay | $10^{-4}$ | $10^{-4}$ |
| Scheduler | Cosine | Cosine |
| Gate bounds | (0.1, 0.9) | (0.1, 0.9) |
| Logit clip $\tau$ | 10 | 10 |

several scores: maximum predictive probability, Dirichlet total evidence $\alpha_0$, predictive entropy, MI (for mixtures), and an energy-based score derived from the learned energy.

# F. Additional Experiments

### F.1. OOD detection performance (AUROC)

Table 8 reports AUROC results for standard ID→OOD benchmarks, using both aleatoric and epistemic uncertainty scores. Here, $A \rightarrow B$ denotes that $A$ is the ID dataset and $B$ the OOD dataset. AUROC is threshold-independent, and higher values indicate better OOD detection performance. Across MNIST-based benchmarks, most recent evidential methods (including ours) saturate near-perfect AUROC, so differences are marginal. On CIFAR-10→SVHN (far-OOD), our methods substantially improve separation: GEM-CORE reaches 94.75/94.36 (alea./epis.), and GEM-FI further boosts epistemic AUROC to 95.09. On CIFAR-10→CIFAR-100 (near-OOD), GEM-FI provides the clearest gains, improving epistemic AUROC from 83.63 (GEM-CORE) to 89.06, suggesting that the feature-informed mixture uncertainty better captures semantic overlap cases where aleatoric cues alone can be insufficient. In contrast, GEM-MIX can underperform on the CIFAR benchmarks (e.g., lower epistemic AUROC on SVHN), which is consistent with mixture assignments becoming less reliable without the feature-informed coupling used by GEM-FI.

### F.2. Precision-Recall Curves

Figure 7 shows Precision–Recall curves for OOD detection (SVHN vs. CIFAR-10) using different uncertainty scores. GEM-FI achieves the highest AUPR across all metrics, indicating a better ranking that preserves precision as recall increases. In particular, mutual information (MI) provides the cleanest separation (93.06%), suggesting that mixture-component disagreement is highly informative for far-OOD

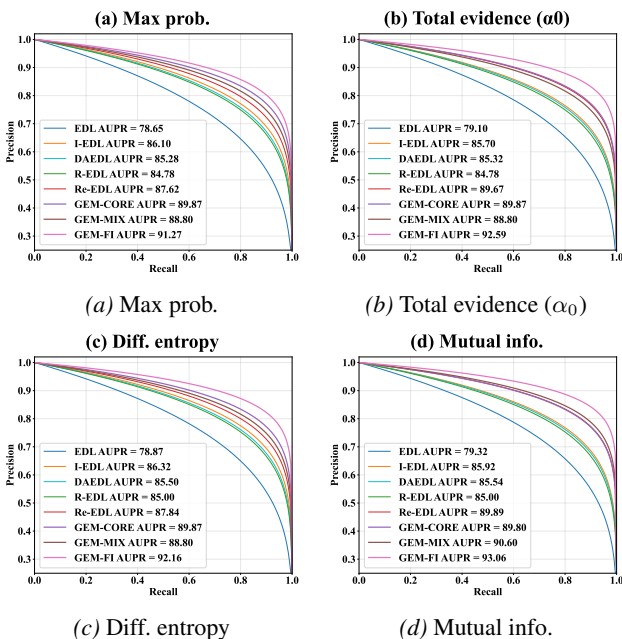

*(a)* Max prob.      *(b)* Total evidence ($\alpha_0$)

*(c)* Diff. entropy     *(d)* Mutual info.

*Figure 7.* Precision–Recall curves for OOD detection with different uncertainty scores.

detection. By contrast, max probability degrades quickly as the threshold is relaxed, consistent with occasional overconfident OOD predictions, while evidence-based and entropy scores are more stable but less selective than MI at higher recall.

### F.3. ROC Curves

Figure 8 shows ROC curves for OOD detection (SVHN vs. CIFAR-10) using the same uncertainty scores. GEM-FI obtains the highest AUROC across metrics, reflecting stronger separability between ID and OOD samples over all thresholds. Among scores, predictive entropy yields the best separation (95.41%), indicating that overall predictive dispersion is a strong cue under far-OOD shifts. MI remains competitive by leveraging component disagreement, whereas max probability and total evidence tend to provide weaker early separation at low false positive rates.

### F.4. Distribution Shift/Corruptions

Although GEM is not specifically designed for corruption robustness, we include a standard corruption stress test for completeness by treating common corruptions as a distribution shift on MNIST-and CIFAR-10-C. We evaluate whether the uncertainty scores can separate clean from corrupted inputs.

Table 9 reports AUPR for detecting corruption-induced shifts using aleatoric uncertainty. For CIFAR-10-C we average over 19 corruption types at each severity level, and we also include MNIST→MNIST-C. As severity increases, detection generally becomes easier (higher AUPR), and the comparison highlights how different evidential variants respond to gradual, in-domain corruptions.

*Table 8.* AUROC for OOD detection using aleatoric and epistemic uncertainty.

| Method | Venue | MNIST → KMNIST | | MNIST → FMNIST | | CIFAR-10 → SVHN | | CIFAR-10 → CIFAR-100 | |
| | | Alea.↑ | Epis.↑ | Alea.↑ | Epis.↑ | Alea.↑ | Epis.↑ | Alea.↑ | Epis.↑ |
| --- | --- | --- | --- | --- | --- | --- | --- | --- | --- |
| DROPOUT | ICML16 | $93.50 \pm 0.10$ | – | $96.10 \pm 0.20$ | – | $50.82 \pm 0.10$ | – | $44.90 \pm 1.00$ | – |
| KL-PN | NeurIPS18 | $92.50 \pm 1.20$ | $92.90 \pm 1.00$ | $97.80 \pm 0.80$ | $97.85 \pm 0.00$ | $43.50 \pm 1.90$ | $42.80 \pm 2.30$ | $60.85 \pm 2.80$ | $61.00 \pm 3.40$ |
| EDL | NeurIPS18 | $96.55 \pm 0.80$ | $95.80 \pm 2.00$ | $97.75 \pm 0.40$ | $97.50 \pm 0.40$ | $81.06 \pm 3.50$ | $81.50 \pm 1.70$ | $80.63 \pm 0.70$ | $80.90 \pm 1.00$ |
| RKL-PN | NeurIPS19 | $60.20 \pm 2.90$ | $53.20 \pm 3.40$ | $77.90 \pm 3.10$ | $71.70 \pm 3.60$ | $53.10 \pm 1.10$ | $48.90 \pm 0.80$ | $54.90 \pm 2.60$ | $54.20 \pm 2.80$ |
| POSTNET | NeurIPS20 | $95.25 \pm 0.20$ | $94.10 \pm 0.30$ | $97.40 \pm 0.20$ | $96.90 \pm 0.20$ | $79.75 \pm 0.20$ | $77.20 \pm 0.40$ | $81.50 \pm 0.80$ | $81.60 \pm 0.80$ |
| *I*-EDL | ICML23 | $97.90 \pm 0.20$ | $97.85 \pm 0.20$ | $98.50 \pm 0.30$ | $98.50 \pm 0.30$ | $86.79 \pm 2.40$ | $86.40 \pm 2.30$ | $82.15 \pm 0.70$ | $81.90 \pm 0.60$ |
| DAEDL | ICML24 | $99.85 \pm 0.00$ | $99.88 \pm 0.00$ | $99.80 \pm 0.00$ | $99.83 \pm 0.00$ | $89.24 \pm 1.40$ | $89.30 \pm 1.40$ | $86.04 \pm 0.10$ | $86.10 \pm 0.10$ |
| R-EDL | ICLR24 | – | $98.20 \pm 0.20$ | – | $99.00 \pm 0.12$ | $86.78 \pm 1.22$ | $86.78 \pm 1.22$ | $85.80 \pm 0.31$ | $85.85 \pm 0.31$ |
| CEDL+ | ESWA25 | $99.80 \pm 0.07$ | $99.82 \pm 0.07$ | $98.70 \pm 0.01$ | $98.68 \pm 0.01$ | $92.50 \pm 0.34$ | $92.55 \pm 0.65$ | $78.90 \pm 0.57$ | $76.50 \pm 0.52$ |
| LTS | MVA25 | $97.70 \pm 0.85$ | $99.90 \pm 0.03$ | $99.50 \pm 0.12$ | $99.70 \pm 0.12$ | $78.10 \pm 0.96$ | $92.00 \pm 0.88$ | $70.70 \pm 0.79$ | $84.80 \pm 0.65$ |
| Re-EDL | TPAMI25 | – | $98.70 \pm 0.28$ | – | $99.55 \pm 0.09$ | $89.72 \pm 0.81$ | $92.19 \pm 1.13$ | $86.67 \pm 0.14$ | $\underline{86.65 \pm 0.14}$ |
| GEM-CORE | | $99.93 \pm 0.01$ | $99.62 \pm 0.42$ | $99.99 \pm 0.00$ | $99.77 \pm 0.30$ | $\mathbf{94.75 \pm 0.24}$ | $94.36 \pm 1.02$ | $\underline{87.30 \pm 0.12}$ | $83.63 \pm 0.12$ |
| GEM-MIX | | $\underline{99.93 \pm 0.02}$ | $\underline{99.93 \pm 0.02}$ | $\underline{99.99 \pm 0.01}$ | $\underline{99.98 \pm 0.02}$ | $93.05 \pm 0.88$ | $81.29 \pm 1.06$ | $83.97 \pm 0.80$ | $74.60 \pm 0.97$ |
| GEM-FI | | $\mathbf{99.94 \pm 0.01}$ | $\mathbf{99.95 \pm 0.01}$ | $\mathbf{99.99 \pm 0.01}$ | $\mathbf{99.99 \pm 0.01}$ | $\underline{93.65 \pm 0.55}$ | $\mathbf{95.09 \pm 0.55}$ | $\mathbf{88.06 \pm 0.06}$ | $\mathbf{89.06 \pm 0.06}$ |

*Table 9.* AUPR scores for distribution-shift detection based on aleatoric uncertainty. For CIFAR-10-C, $C \in 1, 2, 3, 4, 5$ denotes the corruption severity.

| | MNIST → MNIST-C | CIFAR-10 → CIFAR-10-C | | | | |
| | AUPR↑ | $C = 1$ | $C = 2$ | $C = 3$ | $C = 4$ | $C = 5$ |
| --- | --- | --- | --- | --- | --- | --- |
| MAXP | $78.54 \pm 0.30$ | $56.39 \pm 0.70$ | $61.88 \pm 1.10$ | $65.86 \pm 1.30$ | $69.91 \pm 1.50$ | $75.01 \pm 1.80$ |
| EDL | $82.75 \pm 0.80$ | $54.76 \pm 0.30$ | $59.01 \pm 0.40$ | $62.46 \pm 0.50$ | $65.87 \pm 0.60$ | $70.21 \pm 0.80$ |
| *I*-EDL | $86.06 \pm 0.50$ | $56.33 \pm 0.20$ | $61.52 \pm 0.50$ | $65.44 \pm 0.50$ | $69.45 \pm 0.50$ | $74.56 \pm 0.50$ |
| DAEDL | $\mathbf{92.43 \pm 0.30}$ | $\mathbf{57.89 \pm 0.30}$ | $\mathbf{63.23 \pm 0.40}$ | $\mathbf{67.53 \pm 0.40}$ | $\mathbf{72.21 \pm 0.40}$ | $\mathbf{77.74 \pm 0.40}$ |
| GEM-CORE | $87.14 \pm 0.00$ | $54.87 \pm 0.30$ | $58.09 \pm 0.35$ | $60.61 \pm 0.40$ | $63.46 \pm 0.50$ | $68.10 \pm 0.60$ |
| GEM-MIX | $86.42 \pm 0.10$ | $54.45 \pm 0.25$ | $57.54 \pm 0.35$ | $59.84 \pm 0.40$ | $62.48 \pm 0.45$ | $66.98 \pm 0.55$ |
| GEM-FI | $\underline{90.36 \pm 0.10}$ | $55.88 \pm 0.30$ | $59.39 \pm 0.40$ | $61.93 \pm 0.45$ | $65.00 \pm 0.50$ | $69.97 \pm 0.60$ |

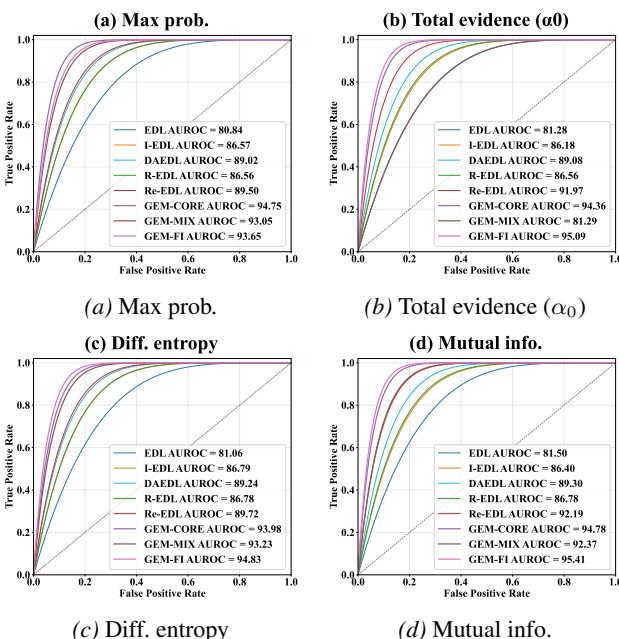

*(a)* Max prob.

*(b)* Total evidence ($\alpha_0$)

*(c)* Diff. entropy

*(d)* Mutual info.

*Figure 8.* ROC curves for OOD detection with different uncertainty scores.

**Discussion.** Corruption benchmarks such as CIFAR-10-C induce in-domain, low-level perturbations while preserving class semantics, making shift detection qualitatively different from semantic OOD settings. In this regime, aleatoric uncertainty (used for the AUPR evaluation in Table 9) need not increase reliably at mild severities: the model can still extract sufficient class evidence and maintain high-confidence predictions even when inputs are corrupted. Since our approach primarily targets epistemic support estimation and ID/OOD separation–i.e., suppressing evidence when representation support is low and stabilizing mixture allocations–sensitivity to gradual within-support corruptions is not explicitly optimized. Accordingly, detection typically improves as corruption severity grows, because larger perturbations more substantially degrade feature support and yield a clearer separation between clean and corrupted samples.

### F.5. Comparison with TS

We compare GEM models against TS and representative EDL-based baselines on CIFAR-10 in the closed-set setting (Table 10). All methods are evaluated using their raw, uncalibrated outputs (i.e., before any post-hoc calibration), except for the TS baseline, which fits a temperature parameter on a held-out validation set. We report: (i) ECE (15 bins) and Brier score ($\times100$) for confidence calibration; (ii) AUPR for misclassification detection; and (iii) mean AUROC averaged over SVHN and CIFAR-100 for OOD detection. Overall, GEM-CORE and GEM-FI achieve competitive (often state-of-the-art) calibration without any post-hoc tuning, while GEM-FI further improves misclassification and OOD detection, indicating that energy-gated evidential learning can yield well-calibrated confidence in a single pass.

*Table 10.* TS comparison on CIFAR-10 (closed-set; pre-calibration unless noted).

| Method | Confidence calibration | | Misclass. detect. | OOD detect. |
|---|---|---|---|---|
| | ECE (15 bins)↓ | Brier (×100)↓ | AUPR↑ | Mean AUROC↑ |
| TS (post-hoc) | **1.06**±**0.10** | 18.44±0.49 | 98.89±0.05 | 82.07±2.23 |
| EDL | 11.56±0.93 | 27.34±0.71 | 98.74±0.07 | 82.32±0.98 |
| $\mathcal{I}$-EDL | 44.35±1.27 | 59.73±1.31 | 98.71±0.11 | 82.01±1.47 |
| DAEDL | 7.22±1.18 | 14.27±0.20 | 99.08±0.00 | 88.19±0.10 |
| R-EDL | 3.47±0.31 | 18.15±0.50 | 98.98±0.05 | 83.73±1.07 |
| Re-EDL | 5.72±0.32 | 14.95±0.47 | 98.81±0.05 | 85.46±1.41 |
| GEM-CORE | 1.94±0.11 | **1.27**±**0.02** | 99.22±0.01 | 88.72±0.21 |
| GEM-MIX | 2.80±0.45 | 6.97±0.03 | 99.43±0.02 | 88.23±0.66 |
| GEM-FI | 2.42±0.04 | 6.81±0.01 | **99.94**±**0.01** | 89.30±0.10 |

*Table 11.* Post-hoc TS on CIFAR-10 for GEM models. Brier: multiclass Brier computed from the final post-gating probabilities, averaged over classes, reported as ×100.

| Model | ECE (%)↓ | | NLL↓ | | Brier (×100)↓ | | $T$ |
|---|---|---|---|---|---|---|---|
| | Before | After | Before | After | Before | After | |
| GEM-CORE | **1.94** | **0.76** | 0.2603 | 0.2553 | **1.27** | **1.23** | 1.18 |
| GEM-MIX | 2.80 | 3.04 | 0.2700 | 0.2697 | 6.97 | 6.96 | 1.01 |
| GEM-FI | 2.42 | 2.70 | **0.2133** | 0.2189 | 6.81 | 6.63 | 0.95 |

### F.6. Effect of Post-hoc TS on GEM

To assess whether GEM models benefit from post-hoc calibration, we apply TS following the standard protocol: we fit a scalar temperature $T$ on a held-out validation set by minimizing the negative log-likelihood, then evaluate calibration metrics before and after scaling (Table 11). We emphasize that TS optimizes NLL and does not necessarily improve ECE.

GEM-CORE learns $T \approx 1.18$, indicating mild overconfidence; applying TS reduces ECE from 1.94% to 0.76% and slightly improves both NLL and Brier. GEM-MIX learns $T \approx 1.01$, consistent with near-calibrated outputs; applying TS leaves NLL and Brier essentially unchanged, and ECE may slightly increase (2.80% → 3.04%). Finally, GEM-FI achieves strong intrinsic calibration (ECE ≈ 2.5%) without post-hoc tuning. With $T \approx 0.95 < 1$, TS slightly sharpens predicted probabilities; while this is appropriate for NLL minimization, it can mildly worsen ECE when calibration is already strong under ECE (2.42% → 2.70%), with Brier remaining similar.

### F.7. Reliability Diagrams and Calibration Sanity Checks

Reliability diagrams assess calibration by plotting empirical accuracy against predicted confidence in fixed-width bins (Guo et al., 2017). A well-calibrated model lies close to the diagonal, while deviations indicate over- or under-confidence.

Figure 9 reports ID reliability diagrams on CIFAR-10 (test set) for GEM-CORE, GEM-MIX, and GEM-FI using the final post-gating probabilities (before any post-hoc calibration). This qualitative view complements Table 11 and confirms that all three variants exhibit low ECE on ID data. To further sanity-check calibration, Figure 10 compares the

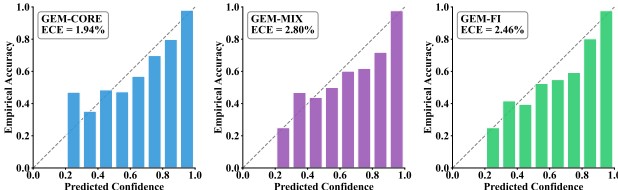

*Figure 9.* Reliability diagrams on CIFAR-10 test (ID). Empirical accuracy vs. predicted confidence (15 equal-width bins) for GEM-CORE, GEM-MIX, and GEM-FI (pre-TS). Reported ECE values match Table 11.

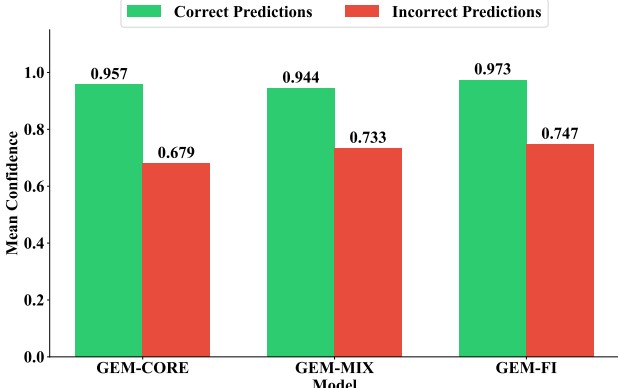

*Figure 10.* Confidence on correct vs. incorrect predictions (CIFAR-10 test, ID). Mean max-confidence for correct and incorrect predictions for GEM-CORE, GEM-MIX, and GEM-FI.

mean max-confidence on correct vs. incorrect predictions. Across all variants, incorrect predictions receive substantially lower confidence than correct predictions, alleviating concerns that unusually low Brier scores could arise from pathological overconfidence.

**Discussion.** Across models, reliability curves remain close to the diagonal on ID data, consistent with the low ECE values in Table 11. Since TS optimizes NLL rather than ECE, applying TS may slightly improve or slightly worsen ECE depending on the model (Table 11). Overall, these diagnostics support that GEM achieves strong ID calibration intrinsically, while maintaining substantially lower confidence on incorrect predictions.

### F.8. Uncertainty Distributions (ID vs OOD)

Figure 11 visualizes how different uncertainty scores separate ID(blue) from OOD(red) samples. We include both digit-domain and natural-image shifts; across pairs, MI provides the clearest ID/OOD separation for GEM-FI, supporting the use of mixture-aware epistemic uncertainty.

### F.9. Entropy–MI Analysis

Figure 12 plots aleatoric uncertainty (entropy) against epistemic uncertainty (MI) for GEM-FI. ID samples concentrate in the low-entropy/low-MI region, while OOD samples typically shift toward higher MI (head disagreement), highlighting the complementarity of the two components.

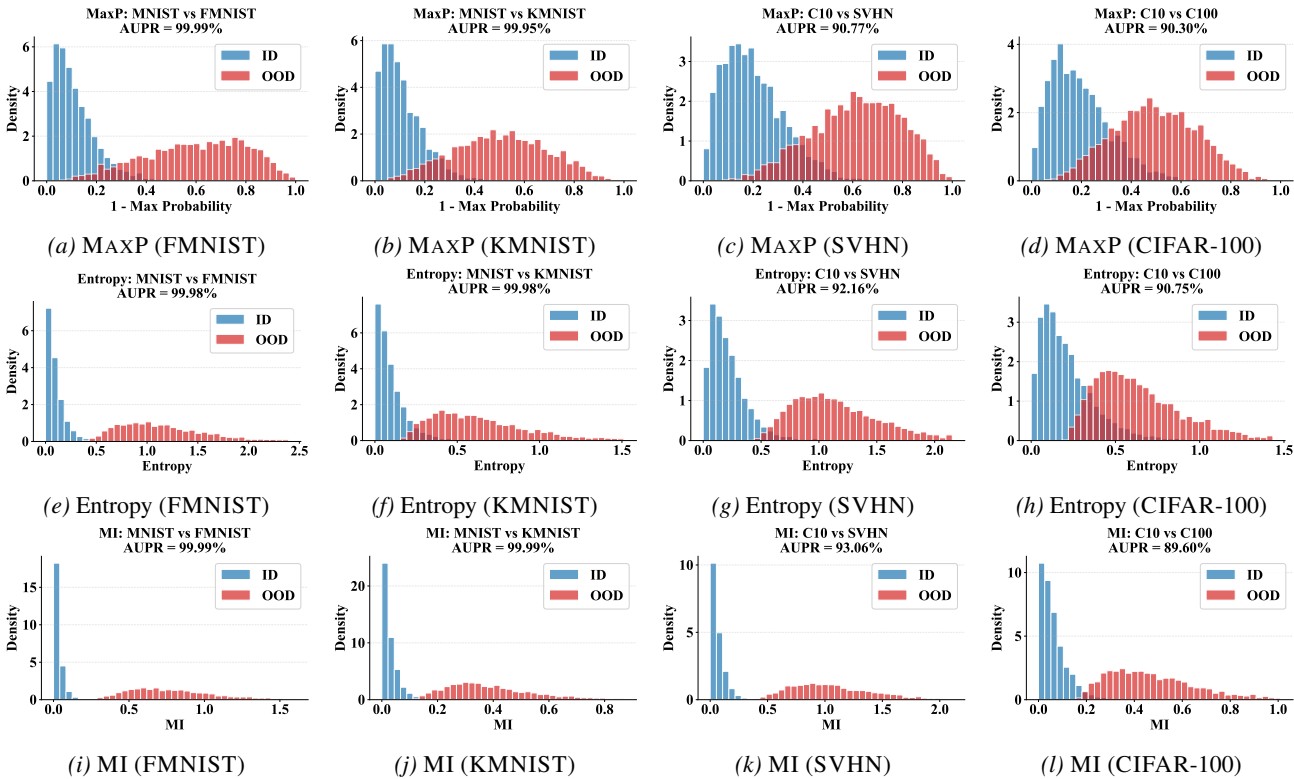

*Figure 11.* Uncertainty distributions measured by (top) maximum probability, (middle) entropy, and (bottom) MI for GEM-FI. Blue = ID samples; Red = OOD samples. MI achieves the best separation, especially for far-OOD pairs.

For mixture models, we decompose predictive uncertainty into aleatoric (entropy) and epistemic (MI) components:

$$\mathrm{MI}(x) = H(\hat{p}(x)) - \sum_{k=1}^{K} \pi_k(x)\, H\big(p^{(k)}(x)\big). \quad (20)$$

High MI indicates between-head disagreement, which is particularly useful for OOD detection.

### F.10. Score Comparison Boxplots

Figure 13 compares common OOD scoring functions–MAXP, predictive entropy, MI, and total evidence $\alpha_0$–across ID(CIFAR-10), near-OOD (CIFAR-100), and far-OOD (SVHN) samples. Well-separated score distributions indicate stronger discriminative uncertainty. In our setting, MI and $\alpha_0$ provide the clearest separation, supporting their use as primary epistemic and evidential signals.

**Discussion.** The boxplots in Figure 13 summarize uncertainty statistics over ID, near-OOD, and far-OOD samples, emphasizing separation between regimes rather than a single scalar score. Compared to baselines, GEM-FI yields higher epistemic indicators (e.g., entropy/MI proxies) on OOD while maintaining lower uncertainty on ID data, suggesting a better trade-off between selectivity and predictive sharpness. Importantly, the improved separation is most visible in the near-OOD setting, where semantic overlap makes OOD detection challenging and overconfidence is common (Hendrycks & Gimpel, 2017).

### F.11. OOD detection performance on dataset-shift benchmarks (AUPR/AUROC)

Table 12 summarizes dataset-shift results for GEM-FI, reporting ID test accuracy and OOD detection metrics computed from multiple uncertainty scores (MAXP, total evidence $\alpha_0$, Energy, predictive Entropy, and mixture-aware MI). For the CIFAR-10→TinyImageNet benchmark, we resize TinyImageNet images from the original $64 \times 64$ to $32 \times 32$ to match the CIFAR-10 input resolution used by our model. This design choice avoids introducing input resolution as an additional confounding factor in the dataset-shift setting, so changes in uncertainty scores and OOD metrics are primarily attributable to distribution shift rather than to input-size or architecture/preprocessing differences. For OOD detection, we treat OOD samples as the positive class and report both AUPR and AUROC (higher is better).

Beyond detection metrics, we include evidential/epistemic diagnostics. In our setting, total evidence $\alpha_0$ and mixture-aware MI are expected to be relatively high on ID inputs and to decrease on OOD inputs, reflecting reduced support and increased epistemic uncertainty under distribution shift. We additionally report Energy and predictive Entropy as complementary uncertainty signals: Energy provides a calibrated separation cue in logit/probability space, whereas Entropy summarizes overall predictive uncertainty. Taken together, improvements in AUPR/AUROC across these scores (alongside the desired ID↑/OOD↓ trends for $\alpha_0$ and

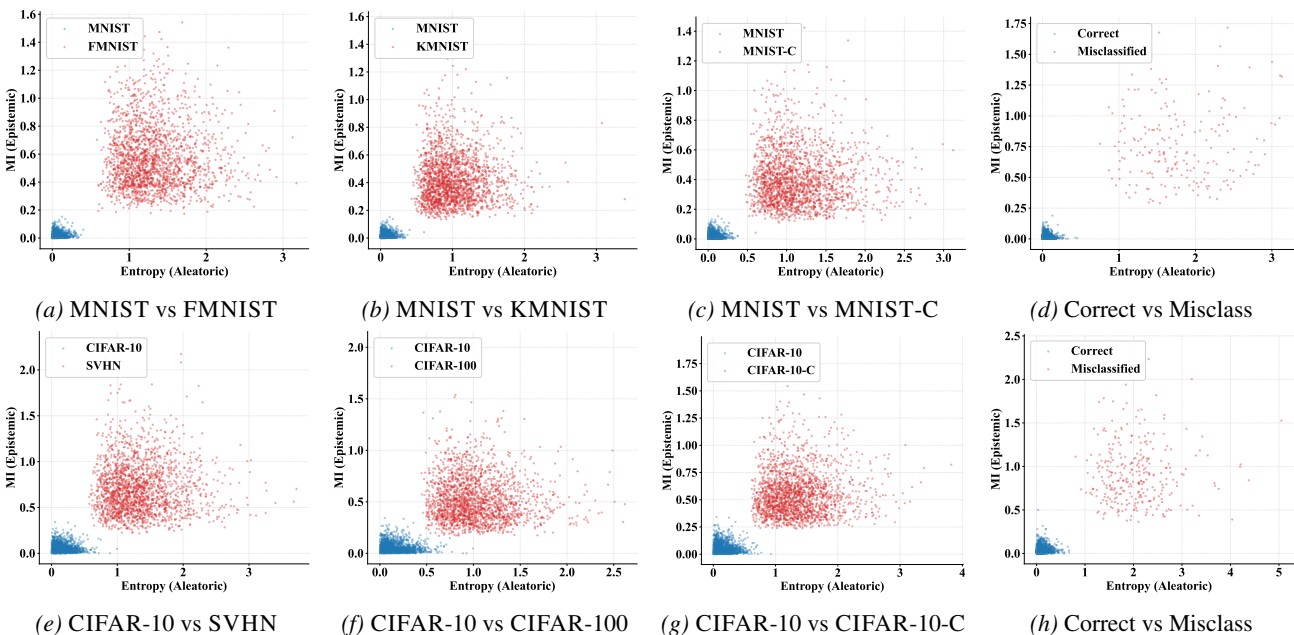

*Figure 12.* Entropy (aleatoric) vs. MI (epistemic) scatter plots for GEM-FI. Top row: MNIST shifts; bottom row: CIFAR-10 shifts. Panels (d) and (h) show Correct vs Misclass for MNIST and CIFAR-10, respectively. ID samples (blue) cluster in the low-entropy, low-MI region, while OOD and corrupted samples (red) exhibit higher values, enabling effective threshold-based OOD detection.

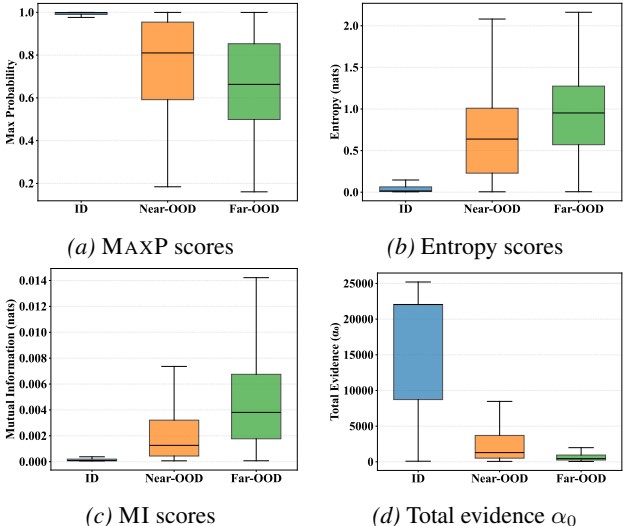

*Figure 13.* Box plots comparing uncertainty scores for ID (CIFAR-10), Near-OOD (CIFAR-100), and Far-OOD (SVHN). MI and $\alpha_0$ show the clearest separation between ID and OOD samples.

MI) provide consistent evidence that GEM-FI yields robust OOD separation without compromising ID accuracy.

# G. Sensitivity Analysis

## G.1. Qualitative Comparison with Baselines

Figure 14 presents a side-by-side comparison of the latent structures learned by GEM-CORE (top row), GEM-MIX (middle row), and GEM-FI (bottom row). The first column shows raw feature embeddings, while the second column depicts the normalized space used for density estimation. The

third column overlays OOD samples (SVHN/CIFAR-100) on ID data (CIFAR-10). Notably, GEM-FI achieves the most compact class clustering and the clearest separation between ID and OOD regions, validating the synergy between the mixture-of-beliefs architecture and Fisher-informed regularization.

## G.2. Effect of $\lambda$ on conflict score

Figure 15 illustrates how the mixing coefficient $\lambda$ controls the relative contribution of inter-class and intra-class conflict in the GEM-FI formulation. We define the conflict score $C$ as a weighted combination of inter-class disagreement (variance of means) and intra-class disagreement (mean of variances), modulated by $\lambda \in [0, 1]$. When $\lambda = 0$, the conflict score depends solely on inter-class disagreement, resulting in vertical gradients dominated by $C_{\text{inter}}$. As $\lambda$ increases, the influence of intra-class conflict becomes more pronounced, leading to smoother diagonal transitions across the conflict landscape. At $\lambda = 1$, the score is fully determined by intra-class inconsistency, encouraging more stable mixture allocations and reducing sensitivity to spurious inter-class fluctuations near decision boundaries.

## G.3. Sensitivity to $\lambda_{FI}$ and $\lambda_{KL}$

To complement the mixture-size analysis, we also examine sensitivity to the Fisher regularization strength $\lambda_{FI}$ and the evidential regularization coefficient $\lambda_{KL}$ on CIFAR-10→SVHN. Across both sweeps, the default setting remains competitive while nearby values yield similar accuracy and OOD separation, indicating that the method is not driven by a narrow hyperparameter sweet spot.

*Table 12.* OOD detection performance of GEM-FI on dataset-shift benchmarks. For each scoring function, reported are the ID test accuracy and the corresponding OOD detection metrics AUPR and AUROC.

| Dataset | Test Acc.↑ | AUPR↑ | | | | | AUROC↑ | | | | |
|---------|-----------|-------|---|--------|---------|-----|--------|---|--------|---------|-----|
| | | MAXP↑ | $\alpha_0$↑ | Energy↑ | Entropy↑ | MI↑ | MAXP↑ | $\alpha_0$↑ | Energy↑ | Entropy↑ | MI↑ |
| MNIST→KMNIST | 98.78 | 99.95 | 99.96 | 99.97 | 99.98 | 99.99 | 99.94 | 99.93 | 99.97 | 99.98 | 99.99 |
| MNIST→FMNIST | 98.78 | 99.99 | 99.99 | 99.99 | 99.98 | 99.99 | 99.99 | 99.99 | 99.98 | 99.98 | 99.99 |
| CIFAR-10→SVHN | 93.75 | 91.27 | 92.59 | 91.35 | 92.16 | 93.06 | 93.65 | 95.09 | 94.03 | 94.83 | 95.41 |
| CIFAR-10→CIFAR-100 | 93.75 | 90.30 | 90.20 | 90.50 | 90.75 | 89.60 | 88.06 | 89.06 | 88.46 | 88.94 | 89.22 |
| CIFAR-10→TinyImageNet | 93.47 | 89.23 | 89.68 | 89.48 | 89.78 | 90.37 | 88.06 | 89.82 | 88.51 | 89.04 | 89.93 |

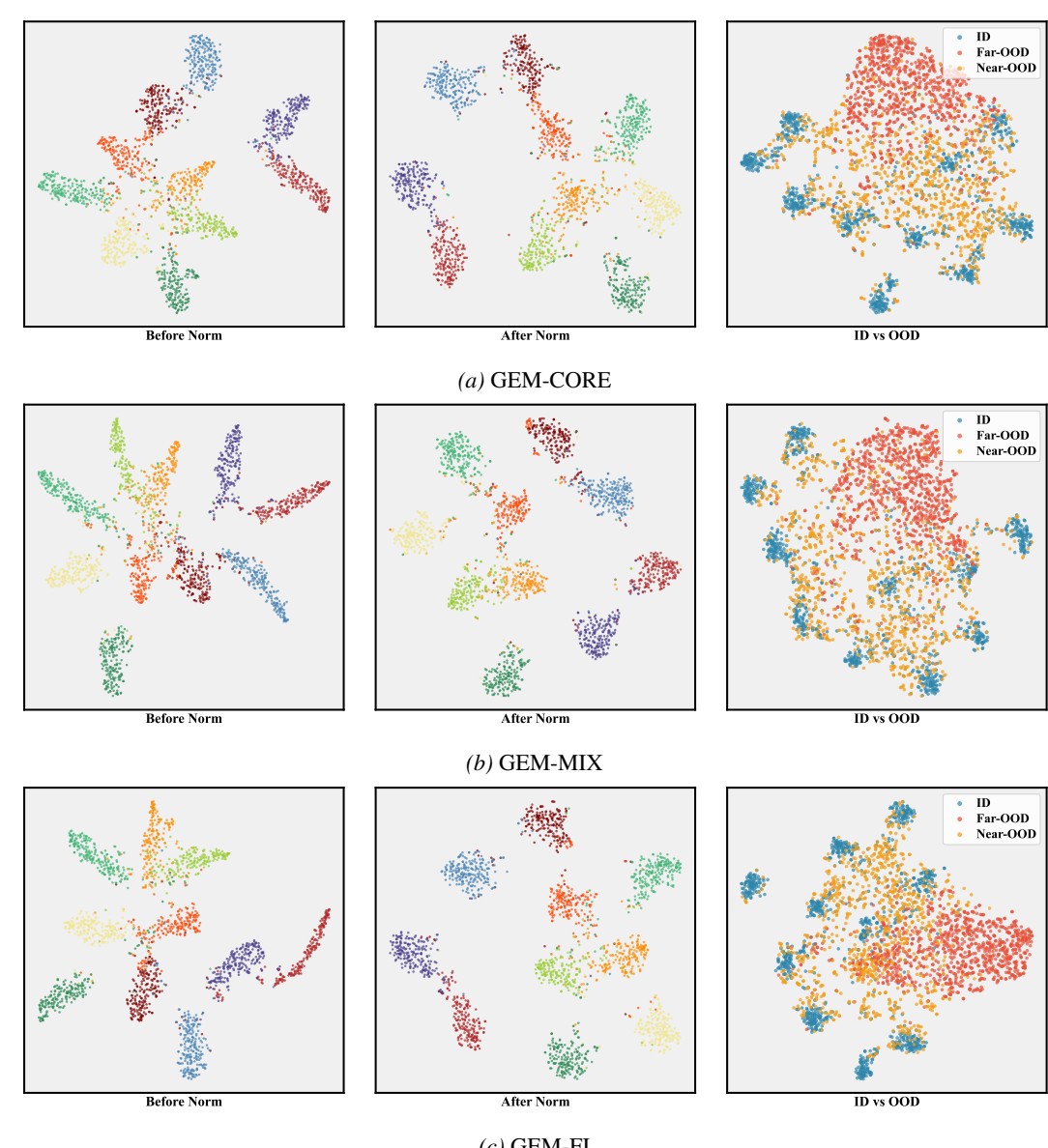

*Figure 14.* Qualitative comparison of feature spaces across methods. Rows: GEM-CORE (top), GEM-MIX (middle), GEM-FI (bottom). Columns: Before Normalization, After Normalization, ID (Blue) vs OOD (Red/Orange).

*Table 13.* Sensitivity to $\lambda_{FI}$ on CIFAR-10→SVHN.

| $\lambda_{FI}$ | Test Acc.↑ | Epis. AUPR↑ |
|------|-----------|-------------|
| 0.01 | $93.68 \pm 0.40$ | $91.43 \pm 0.55$ |
| 0.05 | $93.71 \pm 0.30$ | $92.12 \pm 0.45$ |
| 0.10 | $\mathbf{93.75 \pm 0.36}$ | $\mathbf{92.59 \pm 0.31}$ |
| 0.30 | $93.65 \pm 0.35$ | $92.33 \pm 0.42$ |
| 1.00 | $93.40 \pm 0.50$ | $90.80 \pm 0.80$ |

*Table 14.* Sensitivity to $\lambda_{KL}$ on CIFAR-10→SVHN.

| $\lambda_{KL}$ | Test Acc. | Epis. AUPR |
|------|-----------|------------|
| $10^{-5}$ | $93.60 \pm 0.28$ | $91.9 \pm 0.42$ |
| $10^{-4}$ | $\mathbf{93.75 \pm 0.36}$ | $\mathbf{92.59 \pm 0.31}$ |
| $10^{-3}$ | $93.30 \pm 0.32$ | $92.2 \pm 0.50$ |
| $10^{-2}$ | $92.50 \pm 0.45$ | $91.5 \pm 0.65$ |

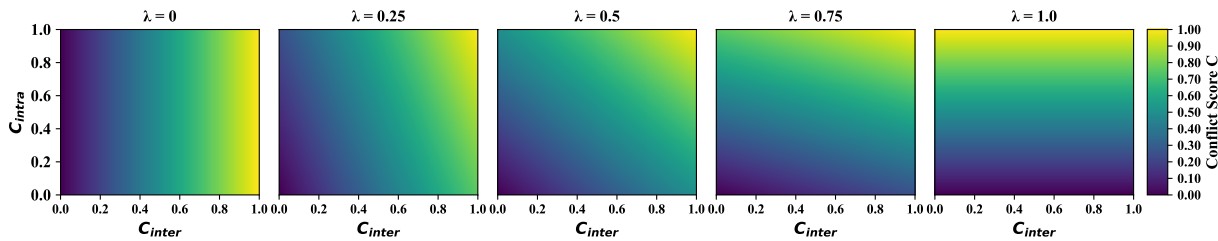

*Figure 15.* Effect of the mixing coefficient $\lambda$ on the conflict score $C$ in GEM-FI. Each panel shows $C$ as a function of inter-class conflict $C_{\text{inter}}$ and intra-class conflict $C_{\text{intra}}$ for a fixed value of $\lambda$ (from left to right: $\lambda \in 0, 0.25, 0.5, 0.75, 1.0$). Larger values of $C$ indicate stronger disagreement between evidential components.

### G.4. Effect of mixture size $K$

To assess the impact of the mixture size in GEM-FI, we vary the number of heads $K \in 3, 4, 5$ and report OOD detection performance together with ID test accuracy on CIFAR-10 (Table 15). We observe that performance is relatively stable across different values of $K$, while $K = 3$ provides the best overall trade-off in these runs, indicating that a small mixture is sufficient to capture meaningful epistemic structure without over-parameterizing the model.

*Table 15.* AUPR scores of OOD detection for GEM-FI with different mixture sizes $K$ on CIFAR-10, based on aleatoric and epistemic uncertainty, along with test accuracy.

| $K$ | Test Acc.↑ | CIFAR-10 → SVHN | | CIFAR-10 → CIFAR-100 | |
|---|---|---|---|---|---|
| | (%) | Alea.↑ | Epis.↑ | Alea.↑ | Epis.↑ |
| 3 | **93.75 ± 0.36** | **91.27 ± 0.29** | **92.59 ± 0.31** | **90.30 ± 0.06** | **90.20 ± 0.06** |
| 4 | 91.71 ± 0.14 | 90.70 ± 0.30 | 92.03 ± 0.21 | 89.04 ± 0.12 | 87.63 ± 0.09 |
| 5 | 92.62 ± 0.20 | 90.62 ± 0.14 | 92.54 ± 0.14 | 90.28 ± 0.08 | 90.15 ± 0.08 |

### G.5. Support-Conditioned Behavior Diagnostics

We analyze how predictive uncertainty, calibration, energy, and accuracy/confidence vary as a function of a feature-space support proxy. Support is measured via $k$NN distance ($k{=}10$) to a CIFAR-10 training feature bank. We report trends for CIFAR-10 (ID), CIFAR-100 (near-OOD), and SVHN (far-OOD).

**Uncertainty vs. support.** Figure 16 shows that predictive uncertainty increases as $k$NN distance grows (i.e., support decreases) across ID, near-OOD, and far-OOD. This monotone rise is visible under both entropy and $1 - \max p$, indicating that samples farther from the CIFAR-10 feature bank are systematically harder and/or less well supported. Compared to the baseline, the proposed variants exhibit a more pronounced uncertainty increase in low-support regions, reflecting more cautious behavior under distribution shift.

**Calibration vs. support.** Figure 17 reports support-conditioned calibration using proper scoring rules (NLL and Brier; ↓ better). Calibration degrades as support decreases, consistent with low-support inputs being more error-prone and shift-prone. Importantly, for matched support levels, the proposed methods generally achieve lower NLL and/or Brier score than the baseline, suggesting that the improved uncer-

tainty behavior is accompanied by better-aligned predictive probabilities rather than merely increased conservatism.

**Accuracy and confidence vs. support.** Figure 18 connects support to performance and confidence. On ID, accuracy decreases with increasing $k$NN distance, reflecting that low-support samples are inherently more challenging. For OOD splits—where accuracy can be less informative due to label-space mismatch—we instead inspect max-confidence: a desirable reliability signature is reduced confidence as support decreases. The proposed variants display a more support-sensitive confidence profile in low-support bins, consistent with improved selective caution under shift.

**Normalized energy vs. support.** Figure 19 plots the learned energy signal against the same support proxy using a normalized scale (e.g., $z$-scoring) to enable comparison across splits and methods. The relationship between energy and the $k$NN proxy is not uniformly monotone and can vary by regime, indicating that energy is not acting as a direct, universal support estimator under this crude proxy. This is expected: the energy head is trained end-to-end as part of the gating/uncertainty mechanism, not to match an explicit density model or support-distance objective. Accordingly, we interpret energy as a learnable control signal whose utility is evidenced indirectly through the support-conditioned reliability improvements in uncertainty, calibration, and confidence (Figs. 16–19).

### G.6. Fair Comparison with VOS

Because VOS is used only in the full GEM-FI pipeline, we also report a matched comparison in which VOS exposure is added to baseline methods and removed from GEM-FI. Table 16 shows that VOS improves all methods to some degree, but does not explain the gains by itself: even without VOS, GEM-FI remains stronger than the VOS-augmented DAEDL and Re-EDL baselines.

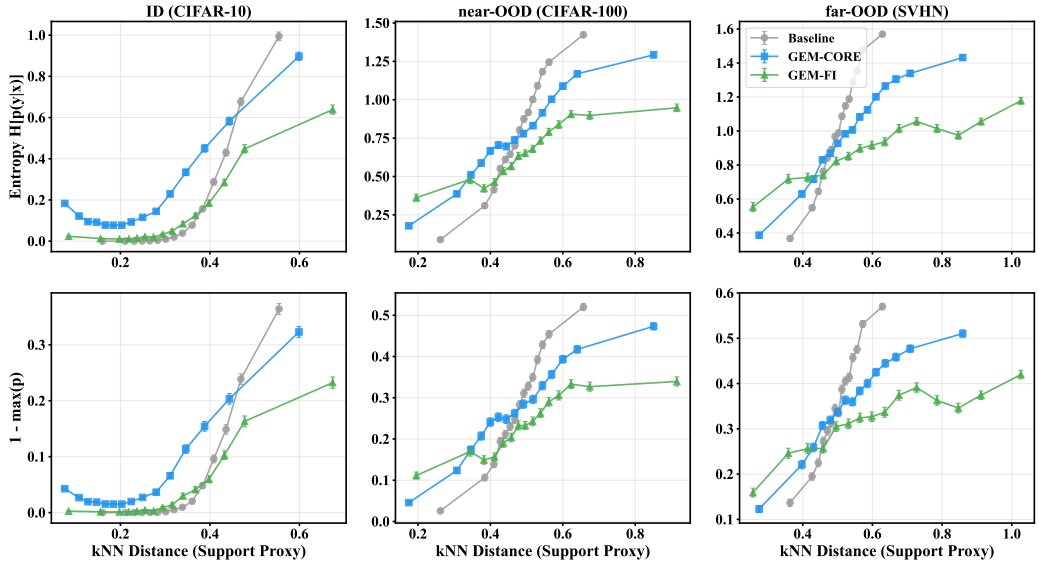

*Figure 16.* Uncertainty vs. support (entropy / $1 - \max p$).

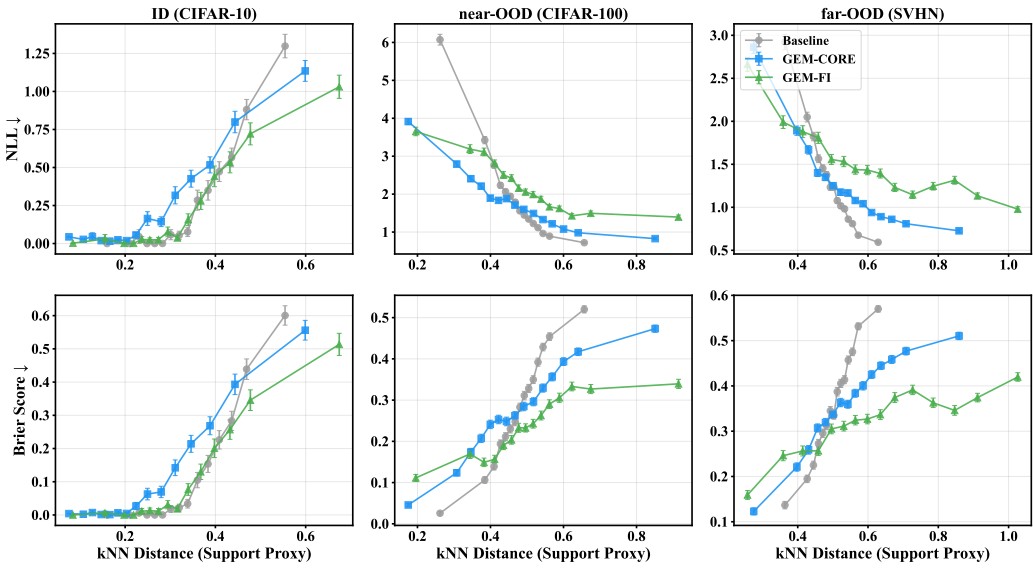

*Figure 17.* Calibration vs. support (NLL, Brier; $\downarrow$ better).

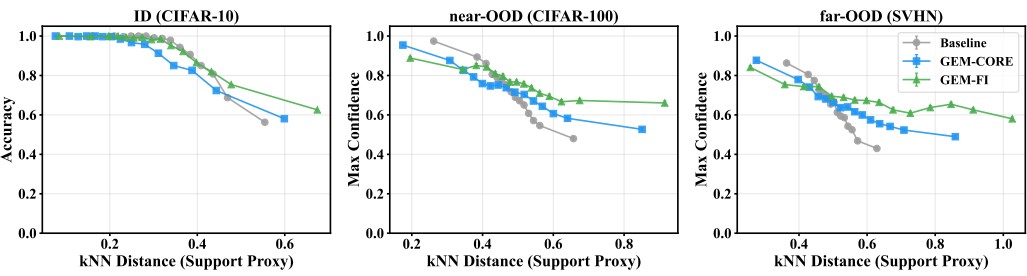

*Figure 18.* Accuracy (ID; $\uparrow$ better) and confidence (OOD; $\downarrow$ better) vs. support.

*Table 16.* Fair comparison with VOS on CIFAR-10$\rightarrow$SVHN. Aleatoric and epistemic columns report AUPR using MaxP and $\alpha_0$/MI-style scores, respectively.

| Method | Alea. AUPR↑ | Epis. AUPR↑ |
|---|---|---|
| DAEDL | $85.50 \pm 1.40$ | $85.54 \pm 1.40$ |
| DAEDL + VOS | $87.12 \pm 1.20$ | $87.34 \pm 1.25$ |
| Re-EDL | $87.84 \pm 0.96$ | $89.89 \pm 1.39$ |
| Re-EDL + VOS | $88.95 \pm 1.00$ | $90.67 \pm 1.10$ |
| GEM-FI (without VOS) | $90.50 \pm 0.30$ | $91.82 \pm 0.32$ |
| GEM-FI (with VOS) | $\mathbf{91.27 \pm 0.29}$ | $\mathbf{92.59 \pm 0.31}$ |

**G.7. Additional VOS and Density-Scaling Ablations**

Table 17 isolates the role of the VOS boundary-sharpening mechanism by replacing it with simple feature-space noise. VOS remains the stronger auxiliary choice, but the core gat-

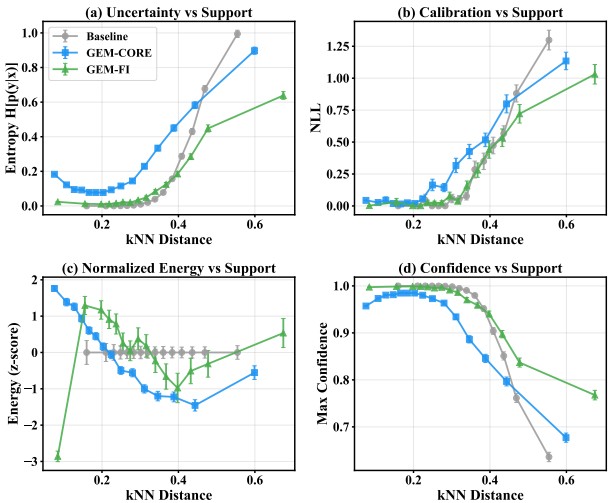

*Figure 19.* Normalized energy vs. support. While the relationship is broadly increasing on average, local non-monotonicity can appear in transition regimes near class boundaries or mixed-support regions; this does not materially affect the bulk OOD detection behavior, where energy still serves as a distance-informed support signal.

ing and mixture design still retain substantial performance without structured synthetic outliers. Table 18 isolates the contribution of the supplementary density scaler $\rho(z)$: most of the gain is retained when $\rho(z) = 1$, while the full model provides a further improvement.

*Table 17.* Effect of replacing VOS with simple feature-space noise on CIFAR-10→SVHN.

| Method | Alea. AUPR↑ | Epis. AUPR↑ |
|---|---|---|
| GEM-FI + VOS | **91.27 ± 0.29** | **92.59 ± 0.31** |
| GEM-FI + feature-space noise | 89.50 ± 0.35 | 89.80 ± 0.38 |

*Table 18.* Effect of removing the supplementary density scaler on CIFAR-10→SVHN.

| Model | Alea. AUPR↑ | Epis. AUPR↑ |
|---|---|---|
| GEM-FI with $\rho(z)$ | **91.27 ± 0.29** | **92.59 ± 0.31** |
| GEM-FI with $\rho(z) = 1$ | 89.50 ± 0.35 | 91.00 ± 0.40 |

### G.8. Energy–Support Correlation Statistics

To complement the qualitative support-conditioned plots in Section G.5, Table 19 reports rank-correlation statistics between learned energy and $k$NN support distance, and Table 20 reports a decile-based summary on CIFAR-10. The average trend is increasing, while local deviations remain concentrated in transition regimes near class boundaries and mixed-support regions.

*Table 19.* Spearman correlation between learned energy and $k$NN support distance.

| Dataset | Spearman $\rho$ | $p$-value |
|---|---|---|
| MNIST | 0.82 ± 0.03 | $< 10^{-6}$ |
| CIFAR-10 | 0.74 ± 0.05 | $< 10^{-6}$ |

*Table 20.* Mean $k$NN support distance across energy deciles on CIFAR-10 test data. Decile 1 denotes lowest energy (highest support), and decile 10 the highest energy regime.

| Energy decile | 1 | 2 | 3 | 4 | 5 | 6 | 7 | 8 | 9 | 10 |
|---|---|---|---|---|---|---|---|---|---|---|
| Mean $k$NN dist. | 4.8 | 5.5 | 6.3 | 7.0 | 7.8 | 8.6 | 9.4 | 10.3 | 11.4 | 12.5 |
| Std. dev. | 1.2 | 1.4 | 1.7 | 1.9 | 2.1 | 2.4 | 2.7 | 3.1 | 3.5 | 3.8 |

### G.9. Head Diversity and Routing Specialization

We further quantify the diversity induced by the mixture architecture. Table 21 shows that head disagreement increases from ID to near-OOD to far-OOD settings, indicating that the mixture captures complementary uncertainty patterns rather than collapsing to identical behavior. Table 22 summarizes router specialization: easy ID cases concentrate most probability mass on one head, while low-support and OOD inputs induce higher routing entropy and softer allocations.

*Table 21.* Head-diversity statistics for GEM-FI with $K = 3$ on CIFAR-10. Pairwise cosine similarity is averaged over all head pairs.

| Split | Pairwise cosine sim. | Disagreement rate |
|---|---|---|
| ID (CIFAR-10 test) | 0.91 ± 0.05 | 9 ± 2% |
| Near-OOD (CIFAR-100) | 0.77 ± 0.09 | 31 ± 3% |
| Far-OOD (SVHN) | 0.61 ± 0.12 | 47 ± 5% |

*Table 22.* Router specialization summary for GEM-FI with $K = 3$ on CIFAR-10. Lower entropy and larger maximum router weight indicate stronger specialization.

| Condition | Head entropy $H(\pi)$ | Mean max router wt. |
|---|---|---|
| Easy ID (high-conf., correct) | 0.22 ± 0.08 | 0.87 ± 0.04 |
| Boundary (low-conf., correct) | 0.75 ± 0.12 | 0.58 ± 0.06 |
| OOD (SVHN) | 1.05 ± 0.09 | 0.42 ± 0.05 |

## H. Failure Modes, Limitations, and Compute Resources

### H.1. Failure Modes

While the proposed method demonstrates strong performance in practice, we observe the following failure modes. First, *head collapse* may occur: without $\pi$-entropy regularization (or with an insufficient coefficient), mixture weights can concentrate on a single head, effectively reducing the model to a single-expert predictor. Second, *confidence lock-in* persists for a small subset of samples: some misclassified inputs remain highly confident due to shortcut features or class-conditional biases learned early in training, and the gate may not sufficiently down-weight these cases. Third, *energy-gate saturation* can happen: very large-magnitude

*Table 23.* Computational overhead comparison on CIFAR-10 with a ResNet-18 backbone.

| Model | Params (M) | Train (s/epoch) | Infer (ms/batch) | Memory (GB) |
|---|---|---|---|---|
| DAEDL | $11.17 \pm 0.00$ | $28.40 \pm 0.5$ | $3.20 \pm 0.1$ | $2.11 \pm 0.0$ |
| GEM-CORE | $11.24 \pm 0.00$ | $30.11 \pm 0.5$ | $3.40 \pm 0.1$ | $2.23 \pm 0.0$ |
| GEM-MIX ($K$=3) | $11.38 \pm 0.00$ | $34.64 \pm 0.6$ | $3.80 \pm 0.1$ | $2.51 \pm 0.1$ |
| GEM-FI ($K$=3) | $11.38 \pm 0.00$ | $42.31 \pm 0.8$ | $3.85 \pm 0.1$ | $3.14 \pm 0.1$ |
| GEM-FI ($K$=5) | $11.52 \pm 0.00$ | $51.70 \pm 1.0$ | $4.22 \pm 0.1$ | $3.86 \pm 0.1$ |

energies may saturate the gating function, reducing sensitivity to intermediate support differences; mild evaluation-time desaturation helps mitigate this effect. Fourth, *expert redundancy* may emerge: multiple heads can converge to similar solutions (especially with limited diversity pressure), which reduces the benefit of maintaining multiple experts and can amplify head collapse. Finally, *training instability under aggressive settings* can arise: large learning rates, very small batch sizes, or strong weight decay may lead to oscillatory mixture weights and brittle gating dynamics.

## H.2. Limitations

Our approach has several practical limitations. First, performance can be sensitive to mixture-related hyperparameters (e.g., entropy strength, number of heads, and gate scaling); poor settings may yield overly sharp routing or overly uniform routing. Second, multi-head evaluation increases inference cost: although the added parameters are small, running multiple heads introduces a modest latency overhead compared to a single-head evidential baseline. Third, the method benefits from stable optimization; in low-precision or memory-constrained regimes, careful tuning (e.g., gradient clipping or conservative schedules) may be required to avoid brittle gating behavior. Fourth, head behaviors are not inherently interpretable: while mixture weights provide some signal, attributing semantic meaning to each head is not guaranteed without additional constraints.

## H.3. Compute Resources

All experiments were conducted on a single NVIDIA RTX 3090 GPU with 32GB system RAM. For CIFAR-10 experiments we used batch size 128; for MNIST we used batch size 50. Training-time overhead is dominated by the FI-related autograd operations used during GEM-FI optimization, while inference remains single-pass and incurs only a modest increase relative to single-head evidential baselines. Detailed parameter counts, training time per epoch, inference latency, and memory usage are reported in Table 23.

