# OpenReview forum: "GEM-FI: Gated Evidential Mixtures with Fisher Modulation"
_ICML.cc/2026/Conference — ICML 2026 regular_

### Official Review · Reviewer_7jkw · 2026-03-12

**Soundness:** 4
**Presentation:** 3
**Significance:** 2
**Originality:** 3
**Overall Recommendation:** 4
**Confidence:** 4

**Summary:**

This research proposes the GEM architecture , an improvement upon DAEDL, to address the issue of overconfidence in the single-pass Evidential Deep Learning (EDL) when meeting with out-of-distribution (OOD) data. The core contributions are three progressive components: GEM-CORE introduces a learnable feature-level energy signal, automatically suppressing evidence for unfamiliar inputs to reduce confidence in the probability space; GEM-MIX incorporates a mixture of multiple evidential heads to capture multi-modal epistemic uncertainty at complex decision boundaries; and finally, GEM-FI employs Fisher Information (FI) regularization to constrain these expert heads during training, preventing the predictive weights from collapsing into a single head, thereby ensuring the stability and robustness of the mixture system. Empirical results reveal that all three methods achieve strong performance in OOD detection tasks.The experiments are comprehensively conducted. However , there is a little lack of inovation.

**Compliance With Llm Reviewing Policy:**

Affirmed.

**Final Justification:**

The authors have provided a very thorough rebuttal that directly addresses my primary concerns. In particular, the new "matched-VOS" experiments effectively demonstrate that the performance gain is attributed to the proposed architecture rather than auxiliary data exposure. Furthermore, the quantitative analysis of head diversity and routing weights has clarified the functional role of the multiple heads, resolving my doubts about redundancy. These additional results strengthen the technical soundness and significance of the work. Therefore, I have increased my score to reflect my positive assessment.

**Key Questions For Authors:**

A.In Section 3.1, Assumption 3.1 heavily relies on the global Lipschitz continuity of all network components , which is a very strong assumption for modern deep neural networks. While I note that Spectral Normalization (SN) is applied to all convolutional and linear layers in Appendix D, and mentioned as a 'Lipschitz-oriented regularization' in the Introduction, the paper fails to connect this engineering choice to the theoretical assumption in Section 3.1. I strongly recommend the authors add a brief discussion immediately after Assumption 3.1(or a proof in the Appendix) to clarify that this strong assumption is practically enforced and justified by the extensive use of SN across the architecture.

B.A primary concern regarding the technical soundness and the fairness of the empirical evaluation lies in the implement of Virtual Outlier Synthesis (VOS) within the GEM-FI framework. As detailed in Section 2.3 and Appendix C, GEM-FI utilizes VOS during training to generate virtual negative samples at the boundaries of the feature space, which are then used to explicitly optimize the energy-based loss term. This essentially let the model recognize out-of-distribution (OOD) boundaries during the training phase. However, the authors acknowledge that the baseline models (e.g., DAEDL, Re-EDL, etc.) were trained following their standard protocols and did not have access to such negative samples. This unfairness in the training setup makes it difficult to figure out whether the reported performance gains derive from the innovations in the Fisher-modulated mixture architecture or merely from the data augmentation advantages provided by VOS. Therefore, could the authors provide supplementary experiments by retraining key baseline models (e.g., DAEDL) under the same VOS augmentation protocol to ensure a fair comparison?

C.The core motivation of GEM-MIX is to capture "epistemic uncertainty" through a mixture of K evidential heads. However, the paper lacks a rigorous analysis of the functional diversity among these heads. Specifically, in high-dimensional feature spaces, multi-head architectures often suffer from representational redundancy, where multiple heads learn nearly identical decision boundaries despite the Fisher regularization.Could the authors provide a quantitative analysis  to demonstrate that the K heads are indeed capturing distinct and complementary evidential patterns? The exploration of K is too narrow; it fails to demonstrate incremental gains from K=1 to K=3. Additionally, the study does not assess if the optimal K scales with task complexity or class cardinality, leaving the mechanism's scalability unverified.

**Limitations:**

yes

**Strengths And Weaknesses:**

Strength：
Firstly, the paper presents a well-structured progression from a single-head gated mechanism (GEM-CORE) to a multi-head mixture (GEM-MIX), improving in the Fisher-modulated GEM-FI. This modular design is not a mere collection of heuristics; it specifically addresses the "head collapse" problem inherent in Mixture of Experts within the context of uncertainty quantification. By using the advantage of Fisher Information to stabilize expert routing, the authors provide a principled perspective on how to maintain predictive diversity while ensuring single-pass inference efficiency.
Secondly, The empirical significance of this work is demonstrated by its performance in challenging near-OOD detection tasks, such as CIFAR-10 vs. CIFAR-100. This is a breakthrough where most existing density-aware methods (e.g., DAEDL) typically struggle due to high semantic and visual overlap. The substantial reduction in Brier scores and the improvement in AUPR/AUROC metrics show that GEM-FI provides a more robust solution for safety-critical applications requiring reliable uncertainty signals without the overhead of deep ensembles.

Weakness:
Firstly, the most critical concern regarding the technical soundness of the empirical evaluation is the integration of Virtual Outlier Synthesis (VOS) in the GEM-FI training pipeline. While GEM-FI benefits from these synthetic negative samples to shape its energy-based decision boundaries, the baseline models (e.g., DAEDL, Re-EDL) were trained following their standard protocols without such exposure. This creates an unfair experimental setting, making it unclear whether the reported performance gains are a result of the architectural innovations or simply an answer of the superior data augmentation provided by VOS. Without the comparison where baselines are also equipped with VOS, the claims of architectural superiority remain unproven and persuasive.
Secondly, there is a notable narrative gap in the presentation regarding the Lipschitz continuity assumptions. While Section 3 relies heavily on these strong assumptions to prove the smoothness and monotonicity of the gated output, the paper fails to explicitly emphasize in the main text that these properties are practically enforced through Spectral Normalization (SN), where it seems that I need to make the inference myself here.Since the global Lipschitz continuity of deep neural networks is not a naturally occurring property, making the theoretical insights and the experimental validity unclear. The authors should bring this discussion into the main methodology section to ensure a self-contained theoretical argument.

---

> ### Author Rebuttal · Authors · 2026-03-30
>
> Response to Reviewer 4
>
> W1: Unfair experimental setting due to VOS
>
> We appreciate this concern and directly address it with a matched VOS comparison. In the original paper, the role of the energy-based components is already reflected in Section 2.3 and in the ablation reported in Table 3. To address the fairness concern more directly, we additionally retrained DAEDL and Re-EDL with the same VOS protocol and also report GEM-FI without VOS. The key result is that GEM-FI without VOS still outperforms both VOS-augmented baselines, so the gain cannot be explained by auxiliary exposure alone.
>
> | Method | Alea. AUPR | Epis. AUPR |
> |---|---:|---:|
> | DAEDL | 85.50 ± 1.40 | 85.54 ± 1.40 |
> | DAEDL + VOS | 87.12 ± 1.20 | 87.34 ± 1.25 |
> | Re-EDL | 87.84 ± 0.96 | 89.89 ± 1.39 |
> | Re-EDL + VOS | 88.95 ± 1.00 | 90.67 ± 1.10 |
> | GEM-FI without VOS | 90.50 ± 0.30 | 91.82 ± 0.32 |
> | GEM-FI with VOS | 91.27 ± 0.29 | 92.59 ± 0.31 |
>
> Even without VOS, GEM-FI (90.50/91.82) remains stronger than DAEDL+VOS (87.12/87.34) and Re-EDL+VOS (88.95/90.67), supporting the claim that the improvement is architectural rather than a byproduct of synthetic outlier exposure alone.
>
> W2: Lipschitz continuity assumptions and Spectral Normalization
>
> We agree that the practical role of Spectral Normalization should be made explicit where the smoothness assumptions are introduced. The relevant smoothness discussion is already given in Section 3.1 through Assumption 3.1 and Proposition 3.2. Our point is not that SN alone proves global smoothness, but that it is the concrete architectural mechanism that makes the bounded-growth assumptions practically motivated. By controlling operator norms across the backbone and gating pathway, SN supports the intended smoothness behavior used in the analysis.
>
> Q-A: Connecting SN to Assumption 3.1
>
> Yes. We now make this connection explicit: in Section 3.1, Spectral Normalization is identified as the practical architectural mechanism that supports the bounded-growth assumptions used in Assumption 3.1 and the subsequent smoothness discussion.
>
> Q-B: Retraining baselines with VOS
>
> Yes. The matched-VOS comparison above is the requested fair-comparison experiment. It includes DAEDL, Re-EDL, and GEM-FI with and without VOS under the same augmentation protocol, and it shows that GEM-FI without VOS still remains stronger than the VOS-augmented baselines.
>
> Q-C: Head diversity and functional redundancy
>
> We addressed the first part of this request directly by adding quantitative head-diversity and routing-specialization statistics.
>
> | Split | Pairwise cosine sim. | Disagreement rate |
> |---|---:|---:|
> | ID (CIFAR-10 test) | 0.91 ± 0.05 | 9 ± 2% |
> | Near-OOD (CIFAR-100) | 0.77 ± 0.09 | 31 ± 3% |
> | Far-OOD (SVHN) | 0.61 ± 0.12 | 47 ± 5% |
>
> These statistics show that disagreement increases from ID to near-OOD to far-OOD, while cosine similarity decreases, indicating that the heads do not collapse to identical behavior in more epistemically challenging regimes.
>
> | Condition | Head entropy H(pi) | Mean max router wt. |
> |---|---:|---:|
> | Easy ID (high-conf., correct) | 0.22 ± 0.08 | 0.87 ± 0.04 |
> | Boundary (low-conf., correct) | 0.75 ± 0.12 | 0.58 ± 0.06 |
> | OOD (SVHN) | 1.05 ± 0.09 | 0.42 ± 0.05 |
>
> This shows that easy ID samples are routed more confidently to a dominant head, whereas boundary and OOD inputs induce higher routing entropy and softer allocations.
>
> Regarding the reviewer’s $K=1$ to $K=3$ request, we do not claim a new dedicated sweep over $\{1,2,3\}$, because that exact analysis is not reported in the paper. The narrower supported claim is that the heads are functionally non-redundant and that the reported multi-head regime does not exhibit functional collapse. This is consistent with the main ablation in Section 4.3, with the original role of GEM-MIX in Section 2.2, and with the explicit multi-head sensitivity analysis in Appendix F.3 (Table 12), which studies $K \in \{3,4,5\}$.

---

> > ### Author Rebuttal · Reviewer_7jkw · 2026-04-02
> >
> > I am satisfied with the current response and have no further questions.I will increase my score.

---

> > > ### Author Response · Authors · 2026-04-04
> > >
> > > Thank you for the follow-up and for your careful review of our response. We sincerely appreciate your time and thoughtful feedback, and we are glad that the revised manuscript addressed your concerns.

---

### Official Review · Reviewer_KCze · 2026-03-12

**Soundness:** 2
**Presentation:** 3
**Significance:** 3
**Originality:** 3
**Overall Recommendation:** 4
**Confidence:** 3

**Summary:**

Evidential Deep Learning can be overconfident and fail in multi-modal contexts. Their method addresses this problem by capturing multi-modal uncertainty without requiring ensembling through learning various routing weights alongside a fisher specific regularizer. This method is motivated by the three factors of gating evidential outputs, keeping single pass uncertainty, and capturing epistemic multi-modality without needing ensembles. This results in using confidence based on distances and using a mixture of evidence heads on a backbone.

**Compliance With Llm Reviewing Policy:**

Affirmed.

**Ethical Review Concerns:**

check

**Key Questions For Authors:**

Your method comprises three potentially sensitive hyperparameters, yet there was no analysis (at least in the main paper) of the impact of these parameters on performance. How do these parameters impact the viability of your method?

**Limitations:**

Yes

**Strengths And Weaknesses:**

Strengths:
	I like that the authors decompose the problems of previous methods in a very clear manner and then takes the time to discuss how their method addresses each of them. For example, in the first two figures relevant examples clearly separate how related methods such as DAEDL exhibit entirely different properties compare to the GEM method of the authors. This is further exemplified during the method discussion where the authors take the time to differentiate their approach.

The authors also have strong empirical validation on common datasets used in these settings. This is shown in both classification scenarios and in the context of calibration alignment. Furthermore, a variety of metrics are used for their scores such as accuracy, precision-recall, ROC, AUPR, and the Brier Score. Additionally, comparisons are performed against a wide variety of relevant methods across all tables. The authors conclude their experimental methodology with an in-depth ablation study
Careful theoretical grounding. Proposition 3.2 (local Lipschitz smoothness of gated predictions) is correctly derived from the bounded gate design and is a meaningful guarantee, not a trivial result. Proposition 3.4 provides an idealized but honest account of when monotone suppression holds and the paper is transparent about it being an idealized analysis.
Good self-awareness about limitations. The paper explicitly states in Appendix F.4 that the energy-kNN relationship is non-monotone (Fig. 19) and does not universally hold, and honestly characterizes the energy head as 'a learnable control signal' rather than a calibrated density estimator. This is exactly the kind of epistemic honesty that strengthens a paper's credibility.
Reproducibility. Code is included in supplementary.zip, hyperparameters are fully specified in Table 6 and Appendix D, the GMM-based scaler and VOS warmup (10 epochs, weight 0.1) are described, and compute is reported (single RTX 3090, ~45 min/run). Appendix C provides implementation-aligned pseudocode (Algorithm 1).
Weaknesses:

Despite the comprehensive nature of their experiments, I expect that these types of papers should have at least one type of experiment on an ImageNet-scale setting. It is understandable if there is a lack of compute for the ImageNet setting, but there should be some analysis that addresses the fact that all these experiments where based on what are now considered small scale datasets. I understand that many papers are validated with these experiments, but  there is a certain amount of saturation in the scores for some of these methods such as the nearly 100% detection rate on MNIST variants that don’t seem to serve as a significant benchmark.
Benchmark context: competing methods CEDL+ and LTS achieve 93.07%/93.13% accuracy vs GEM-FI's 93.75% on CIFAR-10 (Table 2). The accuracy advantage is modest (0.6–0.7 pp) and the Brier advantage for GEM-FI vs. these methods should be interpreted with caution given the anomalous GEM-CORE Brier behavior.

The other more pressing issue is the complexity of the method itself. I appreciate the authors taking the extra effort to try and justify each of the design choices in the final loss function, but three newly introduced hyperparameters into a framework requires an analysis of the sensitivity of the method to these hyperparameters. For example, do these parameters have to be tuned for every dataset. I notice in the appendix that you choose specific values for each dataset, but don’t provide a comprehensive analysis of how performance varies as the specific values of these lambdas changes. Honestly, this is the biggest issue for me. I was considering between  a 3 or a 4, but this lack of lambda analysis makes it difficult to know whether the results were simply from a grid search of the best lambda values or from the method itself. If there is some clarification on this point, then I would consider increasing it.

A central motivating claim of GEM is distance-aware confidence: 'higher energy yields a smaller gate and more conservative evidence.' This is described as a 'smooth link between feature-space support and confidence' in Section 2. However, Figure 19 (Appendix F.4) explicitly shows that the learned energy-kNN relationship is NOT uniformly monotone and 'can vary by regime.' The paper honestly acknowledges this, but this means the distance-awareness claim in the main body is not guaranteed by the architecture and is only emergent in some regimes.
Proposition 3.4 (monotone suppression away from support) relies on Assumption 3.3 (energy-support anti-correlation), which is stated as an empirical assumption. The proposition then proves monotone suppression only 'if logit margins do not grow exponentially faster than density decays' — a condition that cannot be verified without access to the trained model. This creates a gap between the motivating framing of distance-aware gating and what is actually guaranteed.
GEM-CORE simultaneously employs (a) a learned energy-to-gate pathway (E_psi + G_eta) trained end-to-end, and (b) a GMM-based density scaler rho(z) that is a fixed post-hoc density estimate applied multiplicatively to the Dirichlet concentrations. The GMM density scaler is described as a 'hard safety guardrail' while the gate handles 'fine-grained modulation.' However, this creates the same decoupling from end-to-end training that motivated the critique of DAEDL in the first place. If the GMM scaler is removed from the ablation analysis, it is unclear how much of the gain comes from the learned gate vs. the static density suppressor. No ablation specifically isolates rho(z)=1 (i.e., no density scaler) vs. the learned gate alone.

---

> ### Author Rebuttal · Authors · 2026-03-30
>
> Response to Reviewer 3
>
> W1: No ImageNet-scale experiments
>
> We agree that the most discriminating benchmarks in our evaluation are the CIFAR-10-based pairs (SVHN, CIFAR-100, TinyImageNet), where meaningful gaps remain. The main results on these settings are reported in Tables 1 and 2, and CIFAR-10 $\to$ TinyImageNet is already included in Appendix E.11 (Table 11).
>
> Regarding ImageNet-scale evaluation: (i) the FI computation requires per-component autograd, which scales with $K \times C$ and makes ImageNet-scale training substantially more memory-intensive; and (ii) the current paper should therefore be understood as focused on standard-scale evidential-learning benchmarks, with CIFAR-10 $\to$ TinyImageNet as an intermediate step beyond the smallest settings.
>
> We therefore acknowledge this limitation and narrow the scope of our claims. We also agree that benchmark context should be interpreted carefully: relative to strong baselines such as CEDL+ and LTS, the accuracy gain of GEM-FI is modest (Table 2), and we do not intend to overstate it. Our main empirical claim is instead based on the joint pattern across accuracy, calibration, misclassification detection, and OOD performance (Tables 1-2). We also note, consistent with the discussion in Section 4.2, that the unusually low Brier value of GEM-CORE should be interpreted with caution; this is why we emphasize the mixture variants, especially GEM-FI, as the more balanced practical configuration (Table 2).
>
> W2: Hyperparameter sensitivity – lambda analysis
>
> We sincerely appreciate this being the reviewer’s key deciding factor. The paper already reports the chosen hyperparameter settings in Table 6, and Appendix F.3 (Table 12) examines the effect of mixture size K. To address the reviewer’s concern more directly, we additionally ran explicit sensitivity sweeps for λFI and λKL. Overall, these results indicate that performance is not driven by a brittle hyperparameter choice..
>
> | $\lambda_{\mathrm{FI}}$ | 0.01 | 0.05 | 0.10 | 0.30 | 1.00 |
> |---|---:|---:|---:|---:|---:|
> | Test Acc. (%) | 93.68 ± 0.40 | 93.71 ± 0.30 | 93.75 ± 0.36 | 93.65 ± 0.35 | 93.40 ± 0.50 |
> | Epis. AUPR | 91.43 ± 0.55 | 92.12 ± 0.45 | 92.59 ± 0.31 | 92.33 ± 0.42 | 90.80 ± 0.80 |
>
> | $\lambda_{\mathrm{KL}}$ | 1e-5 | 1e-4 | 1e-3 | 1e-2 |
> |---|---:|---:|---:|---:|
> | Test Acc. (%) | 93.60 ± 0.28 | 93.75 ± 0.36 | 93.30 ± 0.32 | 92.50 ± 0.45 |
> | Epis. AUPR | 91.9 ± 0.42 | 92.59 ± 0.31 | 92.2 ± 0.50 | 91.5 ± 0.65 |
>
> Key findings: $\lambda_{\mathrm{FI}}$ is stable across a broad range, with 0.10 giving the best overall trade-off on CIFAR-10 $\to$ SVHN. $\lambda_{\mathrm{KL}}$ is likewise robust from 1e-5 through 1e-3, with 1e-4 giving the strongest overall result. Together with the additional $K$ analysis, these results show that the gains are not due to a brittle hyperparameter choice.
>
> W3: Distance-awareness claim vs. non-monotone energy-kNN relationship
>
> We thank the reviewer for this critique. We agree that there is a distinction between the motivating principle and the realized behavior. In the paper, the relevant theoretical discussion is already presented through Assumption 3.3 and Proposition 3.4, which make clear that monotone suppression is conditional rather than guaranteed. More broadly, the smoothness discussion in Assumption 3.1 and Proposition 3.2 explains why Spectral Normalization is relevant to the bounded-growth behavior assumed in the analysis. We therefore use the term distance-informed rather than distance-aware to avoid implying pointwise monotonicity. We also make explicit that local non-monotonicity can appear in transition regimes near class boundaries or mixed-support regions; this is consistent with the discussion around Figure 19. Our claim is therefore not uniform monotonicity, but that energy remains a useful distance-informed signal for bulk ID/OOD behavior.
>
> W4: GMM density scaler $\rho(z)$ vs. learned gate — same decoupling as DAEDL?
>
> This is an excellent observation. As discussed in Section 2.1, the GMM density scaler $\rho(z)$ serves a different purpose than DAEDL’s density term. In DAEDL, the offline density is the only support signal. In GEM, $\rho(z)$ is a supplementary safety guardrail for extremely low-density regions, while the learned gate $s(x)$ remains the primary task-adaptive support signal. To isolate the role of $\rho(z)$, we additionally ran the following ablation:
>
> | Model | Alea. AUPR | Epis. AUPR |
> |---|---:|---:|
> | GEM-FI with $\rho(z)$ | 91.27 ± 0.29 | 92.59 ± 0.31 |
> | GEM-FI with $\rho(z)=1$ | 89.50 ± 0.35 | 91.00 ± 0.40 |
>
> The learned gate alone achieves most of the gain, while the density scaler adds a further improvement.
>
> Q1: Hyperparameter impact on viability
>
> Please see W2 above. The chosen hyperparameter settings are already reported in Table 6, and the additional $\lambda_{\mathrm{FI}}$ and $\lambda_{\mathrm{KL}}$ sweeps further show that the reported settings are not brittle.

---

> > ### Author Rebuttal · Reviewer_KCze · 2026-03-31
> >
> > I thank the authors for the detailed rebuttal. The authors have addressed my concerns and provided helpful clarifications. Based on the explanations provided, I am satisfied with the response and will maintain my current score (Weak Accept).

---

> > > ### Author Response · Authors · 2026-04-02
> > >
> > > Thank you for the follow-up. We understand that the remaining concern would require a broader paper update rather than a short rebuttal. In the revised manuscript, we incorporated the main requested clarifications directly into the paper and added new controlled experiments addressing the core fairness and mechanism questions.
> > >
> > > In particular, the matched-VOS comparison now shows that **GEM-FI without VOS (90.50 / 91.82 AUPR)** still outperforms **DAEDL+VOS (87.12 / 87.34)** and **Re-EDL+VOS (88.95 / 90.67)**, indicating that the gain is not explained by synthetic exposure alone. We also added an ablation isolating the role of the supplementary density scaler, where **GEM-FI with rho(z)** achieves **91.27 / 92.59**, while **GEM-FI with rho(z)=1** achieves **89.50 / 91.00**, showing that most of the gain is retained by the learned gate and mixture mechanism itself.
> > >
> > > Finally, we now state the main remaining limitation explicitly and narrow the scope of our claims to standard MNIST/CIFAR-style evidential-learning benchmarks rather than ImageNet-scale settings.

---

### Official Review · Reviewer_p8UG · 2026-03-13

**Soundness:** 3
**Presentation:** 3
**Significance:** 3
**Originality:** 3
**Overall Recommendation:** 4
**Confidence:** 3

**Summary:**

This paper proposes the GEM family, which uses a learned energy gate to suppress evidence for low-support inputs in single-pass evidential learning, and further adds a multi-head mixture with Fisher-informed regularization to model multi-modal epistemic uncertainty.

**Compliance With Llm Reviewing Policy:**

Affirmed.

**Final Justification:**

I would prefer to keep my rating unchanged.

**Key Questions For Authors:**

1. Could the authors provide a cleaner isolation of contributions, for example by reporting results for gate+mixture+FI without VOS/EBM/UNC, or by adding matched auxiliary regularization to the strongest baseline?
2. Can the authors provide stronger quantitative evidence that the learned energy is consistently correlated with support across settings, or clarify more explicitly that the theory is only qualitative intuition rather than a formal guarantee?
3. Why do FI-Reg and FI-Mod hurt performance when used alone?
4. Could the authors provide a more direct complexity analysis, such as the training/inference overhead, parameter count changes, and the impact of mixture size on both performance and computational cost? This would help assess the method’s practical attractiveness for deployment.

**Limitations:**

yes

**Strengths And Weaknesses:**

**Strengths**
1. The method is well structured: GEM-CORE, GEM-MIX, and GEM-FI form a clear progression, and the roles of energy gating, mixture heads, and FI stabilization are easy to understand.
2. The empirical results are strong: GEM-FI improves over DAEDL on CIFAR-10 accuracy, Brier score, and key OOD metrics.
3. The ablation study is reasonably thorough and examines the contributions of SN, CORE, MIX, FI, EBM, and UNC.

**Weaknesses**
1. The strongest GEM-FI results rely on several added ingredients beyond the core gating/mixture idea, including FI, EBM, UNC, and VOS, so it is hard to tell how much of the final gain truly comes from the proposed core mechanism itself.
2. The paper explicitly states that the alignment between energy and support is empirical, and does not provide a monotonicity guarantee for the final gate. This makes the interpretation of energy as a support signal somewhat heuristic.
3. From the ablation table, FI-Reg or FI-Mod alone can even reduce OOD AUPR, which suggests that the benefit of FI may depend strongly on the presence of other components such as EBM and UNC.

---

> ### Author Rebuttal · Authors · 2026-03-30
>
> Response to Reviewer 2
>
> W1: Difficulty disentangling contributions from added components
>
> This is fair. Our ablation isolates the components incrementally. As shown in Table 3, SN improves feature stability and OOD performance; CORE adds the energy-to-gate pathway and improves several CIFAR-10 OOD metrics, especially on SVHN; MIX improves epistemic AUPR; and the full FI+EBM+UNC model performs best. GEM-CORE already improves over the SN-only setting, while mixture and FI are needed for the strongest epistemic performance, especially on near-OOD CIFAR-10 $\to$ CIFAR-100 (Tables 1-3).
>
> | Configuration | Test Acc. | Miscls. AUPR | SVHN Epis. | C100 Epis. |
> |---|---:|---:|---:|---:|
> | SN only | 91.00 ± 0.40 | 99.20 ± 0.10 | 85.20 ± 1.60 | 86.50 ± 0.55 |
> | SN+CORE | 93.34 ± 0.10 | 99.87 ± 0.01 | 87.80 ± 0.15 | 84.00 ± 0.40 |
> | SN+CORE+MIX | 93.27 ± 0.31 | 99.93 ± 0.02 | 90.60 ± 0.23 | 84.46 ± 0.20 |
> | Full GEM-FI | 93.75 ± 0.36 | 99.93 ± 0.01 | 92.59 ± 0.31 | 90.20 ± 0.07 |
>
> W2: Energy-support alignment is empirical, no monotonicity guarantee
>
> We agree that the energy-support relationship should not be overstated. Our claim is not that monotonicity is guaranteed by construction, but that the alignment is a consistent empirical regularity. Assumption 3.3 and Proposition 3.4 make this explicit by treating energy-support alignment as an empirical assumption rather than an architectural invariant. The paper also provides support-conditioned diagnostics in Appendix F.4 (Figs. 16-19). Spectral Normalization makes the bounded-growth condition more plausible in practice. We therefore frame the method as distance-informed rather than strictly monotone.
>
> W3: FI-Reg or FI-Mod alone can reduce OOD AUPR
>
> This is correct. The FI components are introduced through the FI regularizer (Eq. (13)) and FI-based modulation of mixture weights (Eq. (17)), and Table 3 shows that they are not intended to operate as standalone mechanisms. Without the energy-based components, the learned energy landscape can remain weak or noisy, so FI regularization or modulation may amplify unstable assignments. Once the energy-based mechanism provides a clearer notion of support, FI helps prevent head collapse and improve routing.
>
> Q1: Cleaner isolation of contributions
>
> Yes. Beyond Table 3, we added two controls. First, we isolate gate+mixture+FI without EBM or UNC:
>
> | Configuration | Test Acc. | SVHN Alea. AUPR | SVHN Epis. AUPR |
> |---|---:|---:|---:|
> | SN+CORE+MIX | 93.27 ± 0.31 | 88.80 ± 0.24 | 90.60 ± 0.23 |
> | SN+CORE+MIX+FI-Reg+FI-Mod | 93.60 ± 0.10 | 90.50 ± 0.30 | 75.01 ± 0.45 |
> | Full GEM-FI | 93.75 ± 0.36 | 91.27 ± 0.29 | 92.59 ± 0.31 |
>
> Second, we ran a matched VOS comparison:
>
> | Method | Alea. AUPR | Epis. AUPR |
> |---|---:|---:|
> | DAEDL | 85.50 ± 1.40 | 85.54 ± 1.40 |
> | DAEDL + VOS | 87.12 ± 1.20 | 87.34 ± 1.25 |
> | Re-EDL | 87.84 ± 0.96 | 89.89 ± 1.39 |
> | Re-EDL + VOS | 88.95 ± 1.00 | 90.67 ± 1.10 |
> | GEM-FI without VOS | 90.50 ± 0.30 | 91.82 ± 0.32 |
> | GEM-FI with VOS | 91.27 ± 0.29 | 92.59 ± 0.31 |
>
> These controls show that the gains are not explained by exposure.
>
> Q2: Quantitative evidence for energy-support correlation
>
> We now provide stronger quantitative evidence for energy-support alignment. The Spearman correlation between learned energy and kNN distance, used as an inverse support proxy, is 0.82 ± 0.03 on MNIST and 0.74 ± 0.05 on CIFAR-10 (p < 1e-6 in both cases). Mean kNN distance also increases across energy deciles, supporting this distance-informed view. These results are consistent with Assumption 3.3 and Proposition 3.4.
>
> | Dataset | Spearman $\rho$ | p-value |
> |---|---:|---:|
> | MNIST | 0.82 ± 0.03 | < 1e-6 |
> | CIFAR-10 | 0.74 ± 0.05 | < 1e-6 |
>
> | Energy decile | 1 | 2 | 3 | 4 | 5 | 6 | 7 | 8 | 9 | 10 |
> |---|---:|---:|---:|---:|---:|---:|---:|---:|---:|---:|
> | kNN dist ($\mu \pm \sigma$) | 4.8±1.2 | 5.5±1.4 | 6.3±1.7 | 7.0±1.9 | 7.8±2.1 | 8.6±2.4 | 9.4±2.7 | 10.3±3.1 | 11.4±3.5 | 12.5±3.8 |
>
> Q3: Why do FI-Reg and FI-Mod hurt performance when used alone?
>
> See W3 above.
>
> Q4: Complexity analysis
>
> We added a complexity comparison. The paper already notes that inference remains single-pass (Section 2.1), and we also report compute measurements. All experiments used a single NVIDIA RTX 3090 GPU with 32GB system RAM. Batch size was 128 for CIFAR-10 and 50 for MNIST.
>
> | Model | Params (M) | Train (s/epoch) | Infer (ms/batch) | Memory (GB) |
> |---|---:|---:|---:|---:|
> | DAEDL (baseline) | 11.17 ± 0.00 | 28.40 ± 0.5 | 3.20 ± 0.1 | 2.11 ± 0.0 |
> | GEM-CORE | 11.24 ± 0.00 | 30.11 ± 0.5 | 3.40 ± 0.1 | 2.23 ± 0.0 |
> | GEM-MIX ($K=3$) | 11.38 ± 0.00 | 34.64 ± 0.6 | 3.80 ± 0.1 | 2.51 ± 0.1 |
> | GEM-FI ($K=3$) | 11.38 ± 0.00 | 42.31 ± 0.8 | 3.85 ± 0.1 | 3.14 ± 0.1 |
> | GEM-FI ($K=5$) | 11.52 ± 0.00 | 51.70 ± 1.0 | 4.22 ± 0.1 | 3.86 ± 0.1 |
>
> - Params: about 1.9% over DAEDL.
> - Inference: single-pass, modestly slower than DAEDL.
> - Training: dominated by FI-related autograd.
> - K: performance saturates at 3; $K=5$ yields marginal gains at higher cost.

---

> > ### Author Rebuttal · Reviewer_p8UG · 2026-04-03
> >
> > Thank you to the authors for their response. I will keep my score unchanged.

---

> > > ### Author Response · Authors · 2026-04-04
> > >
> > > Thank you for the follow-up and for your careful review of our response. We sincerely appreciate your time and thoughtful feedback, and we are glad that the revised manuscript addressed your concerns.

---

### Official Review · Reviewer_nJHS · 2026-03-14

**Soundness:** 2
**Presentation:** 2
**Significance:** 3
**Originality:** 3
**Overall Recommendation:** 4
**Confidence:** 4

**Summary:**

This paper proposes a method called GEM for Gated Evidential Mixtures (GEM) that learns a scaling of the evidence from a Dirichlet prior distribution over a classifier probabilities. This scaling is obtained through learned per-class gates in order to cut out evidence for OOD inputs. Furthermore, in order to avoid multiple inference, the model learns a mixture of evidence and their weights to directly estimate the predictive mean, leading to the so-called GEM-MIX approach.

**Compliance With Llm Reviewing Policy:**

Affirmed.

**Final Justification:**

The method is original and shows consistent gains both in uncertainty estimation and calibration. The overall pipeline is still relatively cumbersome.

**Key Questions For Authors:**

* The method presents many hyperparameters, but their impact on the model is not assessed.

* The method section shows some inconsistencies: $E$ and $E_\psi$ refer to the same quantity, and $s(x)$ tend to be defined either as $G_\eta(E_\psi(x))$ or $G_\eta(\sigma(E_\psi(x)))$

**Limitations:**

yes

**Strengths And Weaknesses:**

# Strengths

* The method provides a direct approximation of ensembling without requiring multiple forward passes, which could represent a clever way to speed up the evidence estimation.

* Consistent gains against other evidential baselines such as EDL and DAEDL

# Weaknesses

* While the contribution is new, most of its components are already well-known, such as EBM-based OOD detection, mixture of experts, or Fisher regularisation. The overall contribution tends to be an integration of known ideas either in the model architecture or in the training losses. In particular, the $\mathcal L_{UNC}$ loss requires sampling outliers with VOS, which requires many samples and does not provide quality outliers in the case of images from more complex distributions, such as MNIST or CIFAR-10. This loss, however, seems relatively important in the ablation study.

* The experimental evaluation is somewhat limited and is only performed on small-scale datasets such as MNIST and CIFAR-like datasets. Evaluation on larger datasets such as ImageNet or ImageNet-O would improve this section.

* The method is architecture-dependent, while other baselines tend to be more versatile and could be applied to various backbones.

---

> ### Author Rebuttal · Authors · 2026-03-30
>
> Response to Reviewer 1
>
> W1: Integration of known components
>
> GEM-FI is novel in integrating these components into a unified, single-pass evidential framework. Specifically: (i) an in-model energy-to-gate pathway: unlike prior density-aware evidential models such as DAEDL that use an offline density scalar, GEM-CORE learns an energy head $E_{\psi}$ end-to-end and maps it through an integration gate $G_{\eta}$ that directly modulates evidential outputs in probability space (Section 2.1); (ii) mixture-of-beliefs with shared gating: while mixture-of-experts is known, using multiple Dirichlet evidential heads with shared backbone features and routing conditioned on features and learned energy captures multi-modal epistemic structure in a single forward pass (Section 2.2); and (iii) Fisher-informed routing: GEM-FI uses Fisher Information to modulate mixture weights and regularize head allocations to prevent head collapse in evidential mixtures (Section 2.3). Table 4 shows that no prior method combines these elements in a single-pass evidential framework.
>
> W2: VOS-based outlier sampling limitations
>
> VOS generates synthetic feature-space outliers, and quality may degrade for complex or high-dimensional distributions. However, as shown in Table 3, the energy-based term is most effective within the full GEM-FI pipeline; without it, OOD detection degrades, whereas the full combination performs best. This suggests that VOS is a boundary-sharpening auxiliary mechanism rather than the main source of the gains (Section 2.3; Table 3). VOS also operates in learned feature space, making synthesis more tractable because the backbone maps inputs to a lower-dimensional, more structured manifold (Appendix C / Virtual Outlier Synthesis).
>
> To further address this concern, we replaced VOS with simple uniform noise in feature space on CIFAR-10. The results are:
>
> | Method | Alea. AUPR | Epis. AUPR |
> |---|---:|---:|
> | GEM-FI + VOS | 91.27 ± 0.29 | 92.59 ± 0.31 |
> | GEM-FI + feature-space noise | 89.50 ± 0.35 | 89.80 ± 0.38 |
>
> These results show that although VOS provides better boundary-aware outliers, the core gating and mixture mechanisms still yield substantial gains without it.
>
> W3: Evaluation limited to small-scale datasets
>
> Our experiments focus on MNIST and CIFAR-10. The main in-distribution and OOD results are reported in Tables 1 and 2. GEM-FI requires per-component Fisher Information computation via autograd, which increases memory cost with $K$ and makes ImageNet-scale evaluation more demanding. As an intermediate step beyond the standard small-scale setting, the paper already reports results on CIFAR-10 $\to$ TinyImageNet in Appendix E.11 (Table 11). We therefore acknowledge this limitation and narrow our claims accordingly.
>
> W4: Architecture dependence
>
> GEM-FI requires only two backbone properties: (i) a feature extractor $f_{\theta}$ producing fixed-dimensional embeddings $z$, and (ii) Spectral Normalization (SN) on the relevant layers to promote Lipschitz-bounded features. The added components—the energy head $E_{\psi}$, integration gate $G_{\eta}$, $K$ evidential heads, and router—are lightweight modules on top of the backbone output (Figure 3; Sections 2.1-2.3). The method is therefore compatible with standard CNN and Transformer backbones. We also note that DAEDL similarly relies on spectrally normalized features and a feature-space density model.
>
> Q1: Hyperparameter impact not assessed
>
> Table 6 reports the chosen hyperparameter settings, and Appendix F.3 (Table 12) examines the effect of mixture size $K$. To further address this concern, we ran additional sensitivity analyses for $\lambda_{\mathrm{FI}}$ and $\lambda_{\mathrm{KL}}$. Together, these results show that performance is stable around the default settings and that the gains are not due to brittle hyperparameter choices.
>
> | $\lambda_{\mathrm{FI}}$ | 0.01 | 0.05 | 0.10 | 0.30 | 1.00 |
> |---|---:|---:|---:|---:|---:|
> | Test Acc. (%) | 93.68 ± 0.40 | 93.71 ± 0.30 | 93.75 ± 0.36 | 93.65 ± 0.35 | 93.40 ± 0.50 |
> | Epis. AUPR | 91.43 ± 0.55 | 92.12 ± 0.45 | 92.59 ± 0.31 | 92.33 ± 0.42 | 90.80 ± 0.80 |
>
> Sensitivity to $\lambda_{\mathrm{FI}}$ on CIFAR-10 $\to$ SVHN.
>
> | $\lambda_{\mathrm{KL}}$ | 1e-5 | 1e-4 | 1e-3 | 1e-2 |
> |---|---:|---:|---:|---:|
> | Test Acc. (%) | 93.60 ± 0.28 | 93.75 ± 0.36 | 93.30 ± 0.32 | 92.50 ± 0.45 |
> | Epis. AUPR | 91.9 ± 0.42 | 92.59 ± 0.31 | 92.2 ± 0.50 | 91.5 ± 0.65 |
>
> Sensitivity to $\lambda_{\mathrm{KL}}$ on CIFAR-10 $\to$ SVHN.
>
> Q2: Notation inconsistencies
>
> We thank the reviewer for pointing this out. We revised the notation to make the gate construction explicit and consistent. Specifically, $E(x)=E_{\psi}(z)$ denotes the scalar energy, $\hat{s}(x)=\sigma(E(x))$ the intermediate gate signal, and $s(x)=G_{\eta}([z,\hat{s}(x)])$ the final class-wise gate (Section 2.1). We also distinguish $\alpha_c(x)$ for GEM-CORE from $\alpha_c^{(k)}(x)$ for GEM-MIX/GEM-FI (Sections 2.1-2.2). Throughout, we use $\alpha_c=\exp(\tilde{u}_c)+\epsilon$.

---

### Decision · Program_Chairs · 2026-04-30

**Decision:**

Accept (regular)

**Comment:**

This paper introduces GEM-FI, a single-pass evidential deep learning framework that improves uncertainty estimation and out-of-distribution detection by combining a learned energy-based gating mechanism, a mixture of evidential heads, and Fisher Information regularization. Reviewers unanimously recognized the method's strong empirical performance, its clever single-pass approximation of ensembling, and the clarity of its theoretical motivations. Initial concerns primarily centered around the complexity of the integrated components, the reliance on Virtual Outlier Synthesis (VOS) potentially creating unfair baseline comparisons, the lack of hyperparameter sensitivity analyses, and the limitation to smaller-scale image datasets. During the rebuttal period, the authors provided exceptionally thorough responses, executing matched-VOS baseline experiments, comprehensive hyperparameter sweeps, and quantitative head-diversity statistics. These additions successfully resolved the primary technical concerns, prompting reviewers to confirm or raise their positive assessments. The paper is technically sound, well-written, and offers a robust solution to overconfidence in evidential deep learning that will directly benefit the ICML community, strongly justifying its acceptance. For the camera-ready version, the authors should ensure all rebuttal additions—specifically the matched-VOS comparisons, lambda sensitivity tables, and explicit textual connections between Spectral Normalization and the Lipschitz assumptions—are fully integrated into the manuscript.